# Approximate Policy Iteration with Bisimulation Metrics

**Mete Kemertas**                                                                    *kemertas@cs.toronto.edu*
*Department of Computer Science, University of Toronto*
*Vector Institute*

**Allan Jepson**                                                                        *jepson@cs.toronto.edu*
*Department of Computer Science, University of Toronto*
*Samsung AI Center, Toronto*

**Reviewed on OpenReview:** *https: // openreview. net/ forum? id= Ii7UeHcOmO*

## Abstract

Bisimulation metrics define a distance measure between states of a Markov decision process (MDP) based on a comparison of reward sequences. Due to this property they provide theoretical guarantees in value function approximation (VFA). In this work we first prove that bisimulation and $\pi$-bisimulation metrics can be defined via a more general class of Sinkhorn distances, which unifies various state similarity metrics used in recent work. Then we describe an approximate policy iteration (API) procedure that uses a bisimulation-based discretization of the state space for VFA and prove asymptotic performance bounds. Next, we bound the difference between $\pi$-bisimulation metrics in terms of the change in the policies themselves. Based on these results, we design an API($\alpha$) procedure that employs conservative policy updates and enjoys better performance bounds than the naive API approach. We discuss how such API procedures map onto practical actor-critic methods that use bisimulation metrics for state representation learning. Lastly, we validate our theoretical results and investigate their practical implications via a controlled empirical analysis based on an implementation of bisimulation-based API for finite MDPs.

## 1 Introduction

Reinforcement learning (RL) algorithms can be broadly grouped into two categories: (i) policy iteration (PI) methods and (ii) policy search methods. The former alternate between learning the value function of the current policy and improving the policy via greedy updates, while the latter directly optimize a performance objective in a feasible set of policies. Learning the value function in large state spaces (e.g., continuous spaces) can be intractable, so value function approximation (VFA) is typically employed for PI in practice. Even with powerful function approximators (e.g., deep neural networks), efficient and generalizable VFA remains an open research problem. State abstraction methods (Li et al., 2006) and state similarity metrics (Lan et al., 2021) take a reductionist approach, aiming to exploit similarities between states to treat them as one, e.g., via state aggregation (Singh et al., 1995; Bertsekas, 2019). In this work we study API via bisimulation metrics with the goal of extending the theory surrounding them, bridging the gap between theory and practice, and improving the stability of API algorithms that rely on them.

Bisimulation metrics measure the *functional* similarity of states by design, comparing only the extent to which reward sequences differ in expectation (Ferns et al., 2004; 2011). Owing to this property, they provide error bounds in VFA, while enabling more efficient state representations. Recent work has tackled challenges in their estimation by introducing $\pi$-bisimulation (Castro, 2020), and employed them for simulated continuous control by constraining the representation space of a neural state encoder (Zhang et al., 2021). Zhang et al. (2021) showed that such constraints promote invariance to background distractors in visual environments, thereby improving sample efficiency in learning. Despite attempts to characterize the trade-offs and con-

vergence properties of practical bisimulation-based RL algorithms (e.g., Kemertas & Aumentado-Armstrong (2021)), PI via bisimulation is still poorly understood.

In this paper we first generalize the definition of bisimulation metrics via $p$-Wasserstein metrics and Sinkhorn distances, and prove their existence for this more general family. This generalization adds theoretical justification to a prior practical modification (Zhang et al., 2021), lifts an assumption on theoretical results concerning VFA via $p$-Wasserstein bisimulation metrics (Kemertas & Aumentado-Armstrong, 2021), connects a recently proposed metric by Castro et al. (2021) to standard bisimulation metrics and allows for fast computation. We then describe an API procedure, which approximates a $\pi$-bisimulation metric at each iteration, and performs state aggregation under it before policy evaluation (PE) and approximate greedy improvement (GI) steps. To make the procedure more efficient, we motivate the use of conservative policy updates, and show that adopting such updates strikes a better trade-off between performance and computational complexity than the naive version. We then conduct a thorough empirical analysis of our theoretical findings to characterize trade-offs posed by various algorithm design choices in terms of asymptotic performance, rate of performance improvement, representation capacity and wall-clock time.

## 2 Background

### 2.1 Setting

Consider a discounted Markov Decision Process (MDP) given by a tuple, $\langle \mathcal{S}, \mathcal{A}, \mathcal{P}, R, \gamma \rangle$: the state and action spaces, transition kernel, reward function and a discount factor $\gamma \in [0, 1)$. For ease of analysis, we assume that the state space $\mathcal{S}$ is *compact*[1] as in Ferns et al. (2011). An agent selects an action, $\boldsymbol{a}_t \in \mathcal{A}$ at each discrete time-step according to a stationary policy $\pi(\boldsymbol{a}_t|\boldsymbol{s}_t)$. The MDP transitions to the next state according to a transition distribution $\mathcal{P}(\boldsymbol{s}_{t+1}|\boldsymbol{s}_t, \boldsymbol{a}_t)$. The distribution over next states when actions are selected according to policy $\pi$ is denoted $\mathcal{P}_\pi(\boldsymbol{s}_{t+1}|\boldsymbol{s}_t)$. With an abuse of notation, we write $\pi(\boldsymbol{s}_t), \mathcal{P}(\boldsymbol{s}_t, \boldsymbol{a}_t)$ and $\mathcal{P}_\pi(\boldsymbol{s}_t)$ for these conditional distributions when appropriate. The agent collects a scalar reward $r_t = R(\boldsymbol{s}_t, \boldsymbol{a}_t)$ from the environment, which is computed via a bounded reward function, $R : \mathcal{S} \times \mathcal{A} \to [0, 1]$. The reward range is selected to simplify analysis, although our theoretical results can be extended to arbitrary bounded reward ranges with ease. The expected immediate reward from choosing an action according to policy $\pi$ in state $\boldsymbol{s}$ is denoted by $R_\pi(\boldsymbol{s}) = \mathbb{E}_{\boldsymbol{a} \sim \pi(\boldsymbol{s})}[R(\boldsymbol{s}, \boldsymbol{a})]$. The agent's discounted return in a given episode is $G = \sum_{t \geq 0} \gamma^t r_t$. We denote by $\mathcal{B}(\mathcal{X})$ the set of real-valued bounded functions over $\mathcal{X}$ and write $V^\pi \in \mathcal{B}(\mathcal{S})$ for the value function of a policy $\pi$, i.e., the fixed point of the Bellman operator $T_\pi : \mathcal{B}(\mathcal{S}) \to \mathcal{B}(\mathcal{S})$, given by the shorthand notation $T_\pi V = R_\pi + \gamma \mathcal{P}_\pi V$. Similarly, $V^*$ denotes the optimal value function, or the fixed-point of the Bellman optimality operator $T : \mathcal{B}(\mathcal{S}) \to \mathcal{B}(\mathcal{S})$, where $TV = \sup_{\pi \in \Pi} T_\pi V$, the supremum taken over the set of stationary policies $\Pi$. For a given function $f$, $\|f\|_\infty$ denotes the supremum (uniform) norm, i.e., $\sup_{x \in \text{dom}(f)} |f(x)|$. We write $\|f\|_{p,\mu}^p = \mathbb{E}_{x \sim \mu}[|f(x)|^p]$ for a distribution $\mu$ supported on $\text{dom}(f)$.

Here, we are interested in state similarity (pseudo) metrics[2] $d : \mathcal{S} \times \mathcal{S} \to [0, \infty)$ to be used to directly aggregate $\mathcal{S}$ or constrain representations of its elements to save space and memory, and to promote efficient learning (e.g., via distraction invariance). In particular, given a state similarity metric $d$ and a threshold $2\epsilon$, one can derive a state abstraction function $\Phi : \mathcal{S} \to \widetilde{\mathcal{S}}$ that satisfies:

$$\Phi(\boldsymbol{s}_i) = \Phi(\boldsymbol{s}_j) \Rightarrow d(\boldsymbol{s}_i, \boldsymbol{s}_j) \leq 2\epsilon. \tag{1}$$

This abstraction $\Phi$ maps a ground MDP $\langle \mathcal{S}, \mathcal{A}, \mathcal{P}, R, \gamma \rangle$ to its abstract version $\langle \widetilde{\mathcal{S}}, \mathcal{A}, \widetilde{\mathcal{P}}, \widetilde{R}, \gamma \rangle$, where $\widetilde{\mathcal{P}}$ and $\widetilde{R}$ are defined as per-partition weighted averages. In particular, let $B_\Phi(\boldsymbol{s}) = \{\boldsymbol{z} \in \mathcal{S} \mid \Phi(\boldsymbol{s}) = \Phi(\boldsymbol{z})\}$ be the set of states that are in the same partition as $\boldsymbol{s}$. Given an arbitrary non-negative measure $\xi$ that assigns

---

[1]All finite discrete spaces are compact. A continuous space is compact if and only if it is totally bounded and complete.
[2]For brevity, we drop "pseudo" in the following.

positive measure $\xi(B_\Phi(\boldsymbol{s})) > 0$ to all partitions $B_\Phi(\boldsymbol{s})$, we write (Li et al., 2006; Ferns et al., 2011):

$$
\begin{aligned}
\widetilde{R}(\Phi(\boldsymbol{s}), \boldsymbol{a}) &:= \frac{1}{\xi(B_\Phi(\boldsymbol{s}))} \int_{\boldsymbol{z} \in B_\Phi(\boldsymbol{s})} R(\boldsymbol{z}, \boldsymbol{a}) d\xi(\boldsymbol{z}), \\
\widetilde{\mathcal{P}}(\Phi(\boldsymbol{s}')|\Phi(\boldsymbol{s}), \boldsymbol{a}) &:= \frac{1}{\xi(B_\Phi(\boldsymbol{s}))} \int_{\boldsymbol{z} \in B_\Phi(\boldsymbol{s})} \mathcal{P}(B_\Phi(\boldsymbol{s}')|\boldsymbol{z}, \boldsymbol{a}) d\xi(\boldsymbol{z}).
\end{aligned}
\tag{2}
$$

Next, we discuss bisimulation metrics, which provide guarantees in approximating value functions over the ground MDP using value functions over abstract MDPs derived via (1). Further background and references on state aggregation methods are provided in Appendix B.

## 2.2 Bisimulation Metrics for Continuous MDPs

To provide VFA guarantees, Ferns et al. (2011) defined the following bisimulation metric for continuous MDPs as a weighted sum of the difference between immediate rewards obtained from respective states and a future-looking recursive term based on the 1-Wasserstein distance (see (8) with $p = 1, \lambda = 0$ for a definition).

**Definition 2.1** (Bisimulation metric for continuous MDPs, Thm. 3.12 of (Ferns et al., 2011))**.** Let $\mathfrak{met}(\mathcal{S})$ be the set of bounded pseudo-metrics over a compact $\mathcal{S}$. Given $c_R \in [0, \infty)$ and $c_T \in (0, 1)$, the following mapping $\mathcal{F} : \mathfrak{met}(\mathcal{S}) \to \mathfrak{met}(\mathcal{S})$ has a unique fixed-point $d^\sim$ called the bisimulation metric:

$$
\mathcal{F}(d)(\boldsymbol{s}_i, \boldsymbol{s}_j) = \max_{\boldsymbol{a} \in \mathcal{A}} c_R |R(\boldsymbol{s}_i, \boldsymbol{a}) - R(\boldsymbol{s}_j, \boldsymbol{a})| + c_T W_1(d)(\mathcal{P}(\boldsymbol{s}_i, \boldsymbol{a}), \mathcal{P}(\boldsymbol{s}_j, \boldsymbol{a})).
\tag{3}
$$

The existence proof applies to compact state spaces via the Banach fixed-point theorem (Ferns et al., 2011). A special case of this metric for finite MDPs was also outlined previously by Ferns et al. (2004). Ferns et al. (2011) showed that this formulation ensures a connection to optimal value functions. In particular, whenever $c_T \geq \gamma$, $V^*$ is $c_R^{-1}$-Lipschitz under the bisimulation metric, i.e., $c_R|V^*(\boldsymbol{s}_i) - V^*(\boldsymbol{s}_j)| \leq d^\sim(\boldsymbol{s}_i, \boldsymbol{s}_j)$. The Lipschitz continuity of $V^*$ with respect to $d^\sim$ results in the VFA guarantee that given a state abstraction $\Phi$ derived via $d^\sim$ as in (1), whenever $c_T \in [\gamma, 1)$:

$$
\|V^* - \widetilde{V}_\Phi^*\|_\infty \ \leq \ \frac{2\epsilon}{c_R(1-\gamma)},
\tag{4}
$$

where $\widetilde{V}_\Phi^*(\boldsymbol{s}) = \widetilde{V}^*(\Phi(\boldsymbol{s}))$. In words, whenever the bisimulation metric places as much weight on future distances as the value function places on future rewards, $\epsilon$-aggregation[3] under the bisimulation metric yields an abstract MDP, which has an optimal value function that is close to that of the ground MDP (Ferns et al., 2011). Thus, given knowledge of the bisimulation metric, one can reduce an MDP with a possibly continuous state space to a finite MDP, which can be solved easily via regular (exact) PI, and have confidence that the solution is approximately optimal with worst-case error given as a linear function of the aggregation radius $\epsilon$, which determines the granularity of the partitioning. Given that exact PI for finite MDPs converges in $\mathcal{O}\left(\frac{|\mathcal{S}||\mathcal{A}|}{1-\gamma} \log(\frac{1}{1-\gamma})\right)$ steps (Scherrer, 2013), if substantial reductions in the size of the state space are possible (e.g., due to the presence of distractors) such that $|\mathcal{S}| \gg |\widetilde{\mathcal{S}}|$, one can pre-compute bisimulation-based abstractions to find a near-optimal policy much more quickly. However, computing the metric itself exactly can be costly; for example, fixed-point iteration requires $\mathcal{O}\left(\frac{\log(\mathcal{E})}{\log(c_T)}|\mathcal{S}|^5|\mathcal{A}| \log|\mathcal{S}|\right)$ steps in the worst-case to find an approximate bisimulation metric $\widehat{d}$ such that $\|\widehat{d} - d^\sim\|_\infty \leq \mathcal{E}$ (Ferns et al., 2006). Assuming that $\Phi$ can be computed at a low cost, a naive combination of bisimulation-based partitioning followed by PI over the reduced MDP yields a complexity of $\mathcal{O}\left(\frac{\log(\mathcal{E})}{\log(c_T)}|\mathcal{S}|^5|\mathcal{A}| \log|\mathcal{S}| + \frac{|\widetilde{\mathcal{S}}||\mathcal{A}|}{1-\gamma} \log(\frac{1}{1-\gamma})\right)$, which need not be superior to directly applying PI on a finite ground MDP. Further, fixed-point iteration cannot be used trivially over continuous state spaces so that function approximation needs to be adopted. Hence, fast approximations of bisimulation metrics are necessary in practice to amortize the cost of metric learning and enable usage in continuous MDPs or large finite MDPs. Indeed, in Sec. 3.1 we consider fast approximation of Wasserstein distances, which is a major bottleneck that prior work attempted to overcome or circumvent via assumptions (Ferns et al., 2006; Castro, 2020; Zhang et al., 2021; Castro et al., 2021).

---

[3]An $\epsilon$-aggreagated state space is any partitioning of $\mathcal{S}$ that permits a maximum partition radius of $\epsilon$ under a metric $d$.

### 2.3 $\pi$-bisimulation Metrics

While bisimulation metrics are useful for approximating $V^*$ of a large MDP, they can be difficult to compute for large (e.g., continuous) action spaces due to the max operation in (3). Secondly, the max operator is inherently pessimistic in assigning a notion of similarity to states. Castro (2020) highlighted these issues and proposed $\pi$-bisimulation metrics to address them.

**Definition 2.2** ($\pi$-bisimulation metric (Castro, 2020))**.** Given a fixed policy $\pi$, the following mapping $\mathcal{F} : \mathfrak{met}(\mathcal{S}) \rightarrow \mathfrak{met}(\mathcal{S})$ has a unique fixed-point $d_\pi^\sim$ called the $\pi$-bisimulation metric:[4]

$$\mathcal{F}_\pi(d)(\boldsymbol{s}_i, \boldsymbol{s}_j) \coloneqq c_R|R_\pi(\boldsymbol{s}_i) - R_\pi(\boldsymbol{s}_j)| + c_T W_1(d)(\mathcal{P}_\pi(\boldsymbol{s}_i), \mathcal{P}_\pi(\boldsymbol{s}_j)). \tag{5}$$

An approach to learning $d_\pi^\sim$ via stochastic approximation with replay buffer samples was presented; the approach reduces the complexity of metric learning by a factor of $|\mathcal{A}|$ for finite MDPs. Castro (2020) also showed that the value function $V^\pi$ of a policy is similarly 1-Lipschitz under the $\pi$-bisimulation metric when $c_R = 1$, i.e., $|V^\pi(\boldsymbol{s}_i) - V^\pi(\boldsymbol{s}_j)| \leq d_\pi^\sim(\boldsymbol{s}_i, \boldsymbol{s}_j)$. Recently, Kemertas & Aumentado-Armstrong (2021) assumed that $\mathcal{F}_\pi$ has a unique fixed-point if defined via an arbitrary $p$-Wasserstein metric with $p \geq 1$ instead of the 1-Wasserstein metric specifically. Then, given an abstraction $\Phi$ derived via $d_\pi^\sim$, for any $c_T \in [\gamma, 1)$ and $p \geq 1$,

$$\|V^\pi - \widetilde{V}_\Phi^\pi\|_\infty \ \leq \ \frac{2\epsilon}{c_R(1 - \gamma)}, \tag{6}$$

where $\widetilde{V}^\pi = \widetilde{R}_\pi + \gamma \widetilde{\mathcal{P}}_\pi \widetilde{V}^\pi$. Similarly to (2), $\widetilde{R}_\pi$ and $\widetilde{\mathcal{P}}_\pi$ were defined as per-partition weighted averages of $R_\pi$ and $\mathcal{P}_\pi$ respectively (Kemertas & Aumentado-Armstrong, 2021). In the next section, we use (6) to construct API algorithms with performance bounds.

## 3 Theoretical Analysis

In this section, we first prove that bisimulation metrics can be defined via a more general class of statistical distances including $p$-Wasserstein metrics and Sinkhorn distances (Cuturi, 2013), which can be used to compute upper bounds on the 1-Wasserstein metric at improved complexity. Based on these results, we will derive a feasible API procedure with bounded error to optimality. The procedure performs alternating updates to a sequence of policies $\pi_k$ and approximations of corresponding sequence of metrics $d_{\pi_k}^\sim$. Unlike Zhang et al. (2021), we do not assume a continuously improving policy to argue for convergence. Rather, we leave the possibility of *policy oscillation* open (unlike exact PI, approximate PI is not guaranteed to converge (Bertsekas & Tsitsiklis, 1996)), but provide asymptotic bounds on optimality. Next we show that restricting the size of policy updates renders such procedures more stable, making a case for the use of conservative policy updates in the context of bisimulation. To further this point, we outline an API($\alpha$) algorithm that bounds the policy update size, and consequently enjoys better performance bounds than the naive API algorithm. We conclude the section by discussing the connections between our theoretical setting and practical algorithms used for larger-scale problems. All proofs are relegated to the Appendix for space.

### 3.1 On the Use of Optimal Transport for Bisimulation Metrics

In prior theoretical results, bisimulation metrics were defined via the 1-Wasserstein metric (i.e., the Kantorovich metric) (Ferns et al., 2004; 2011; Castro, 2020). $p$-Wasserstein distance computation between a pair of distributions over a finite space $\mathcal{S}$ has worst-case complexity $\mathcal{O}(|\mathcal{S}|^3 \log |\mathcal{S}|)$ (Orlin, 1988). This makes the usage of the 1-Wasserstein metric a major obstacle for practical use since for a single fixed-point update it is computed $|\mathcal{S}|^2|\mathcal{A}|$ and $|\mathcal{S}|^2$ times for (3) and (5) respectively. To circumvent this problem in empirical studies, Castro (2020) assumed deterministic dynamics.[5] Similarly, Zhang et al. (2021) assumed the dynamics can be modelled as Gaussians over a latent space and successfully used a 2-Wasserstein metric to exploit the closed-form of the $W_2$ distance between Gaussians (Olkin & Pukelsheim, 1982), albeit without

---

[4]Castro (2020) originally defined the metric with $c_R = 1$ and $c_T = \gamma$.

[5]The Wasserstein distance between two delta distributions is simply the distance between the two points (Villani, 2008).

theoretical justification. Here, we show that $p$-Wasserstein distances can indeed be used safely and thereby lift the assumption made by Kemertas & Aumentado-Armstrong (2021) to prove (6) for arbitrary $p \geq 1$.

We further show that Sinkhorn distances, which bound Wasserstein distances above via entropic regularization, can also be used. The practical advantages of using Sinkhorn distances are three-fold: (i) owing to a strictly convex optimization objective, the Sinkhorn-Knopp algorithm (Sinkhorn & Knopp, 1967) can be used to compute them in $\mathcal{O}(|\mathcal{S}|^2 \log |\mathcal{S}|)$ time (Altschuler et al., 2017; Dvurechensky et al., 2018), (ii) unlike standard Wasserstein distance solvers this computation can be massively parallelized on GPUs (Cuturi, 2013), and (iii) between fixed-point iterations of $\mathcal{F}_\pi$ one can easily save Sinkhorn potentials to warm-start the Sinkhorn-Knopp algorithm at an overall memory cost of $\mathcal{O}(|\mathcal{S}|^3)$. Now, we define primal and dual Sinkhorn distances with $p \geq 1$ in preparation of a generalized definition of bisimulation metrics.

**Definition 3.1** (($p,\zeta$)- and ($p,\lambda$)-Sinkhorn distances). Let $d : \mathcal{X} \times \mathcal{X} \to [0, \infty)$ be a distance function and $\Omega$ the set of all joint distributions over $\mathcal{X} \times \mathcal{X}$ with marginals $\mu_1, \mu_2 \in \mathcal{P}_p(\mathcal{X})$, where $\mathcal{P}_p(\mathcal{X})$ denotes the set of probability measures with bounded moments of order $p$ on $\mathcal{X}$. Given the product of marginals $\mu_1 \otimes \mu_2$ (Genevay et al., 2016) and $p \geq 1, \zeta \geq 0$, we call the following primal form ($p,\zeta$)-Sinkhorn distances:

$$W_p^\zeta(d)(\mu_1, \mu_2) = \min_{\omega \in \Omega(\zeta)} \|d\|_{p,\omega}, \text{ where } \Omega(\zeta) = \{\omega \in \Omega \mid D_{\mathrm{KL}}(\omega \, \| \, \mu_1 \otimes \mu_2) \leq \frac{1}{\zeta}\}. \tag{7}$$

The dual form is given by the Lagrangian of (7) with $\lambda \geq 0$:

$$W_p^\lambda(d)(\mu_1, \mu_2) = \|d\|_{p,\omega^*}, \text{ where } \omega^* = \arg\min_{\omega \in \Omega} \|d\|_{p,\omega}^p - \lambda \mathcal{H}(\omega), \tag{8}$$

where $\mathcal{H}$ denotes Shannon entropy. To each $\zeta$ and pair of distributions $(\mu_1, \mu_2)$ corresponds a $\lambda$ such that $W_p^\lambda(d)(\mu_1, \mu_2) = W_p^\zeta(d)(\mu_1, \mu_2)$ (Cuturi, 2013). While $\lambda = 0$ recovers $p$-Wasserstein distances as a special case where $\zeta$ is sufficiently small, $\lambda > 0$ renders the objective of the dual form strictly convex. Consequently, when computing $W_p^\lambda(d)(\mu_1, \mu_2)$ one can use the Sinkhorn-Knopp algorithm, which converges in fewer iterations for higher $\lambda$ (Cuturi, 2013), albeit at the expense of weaker upper bounds on $W_p(d)(\mu_1, \mu_2)$.

**Lemma 3.2** (A ($p,\zeta$)-Sinkhorn distance bound). *Given metrics $d$ and $d'$, for all $p \geq 1$ and $\zeta \geq 0$,*

$$\left| W_p^\zeta(d)(\mu_1, \mu_2) - W_p^\zeta(d')(\mu_1, \mu_2) \right| \leq \|d - d'\|_\infty. \tag{9}$$

**Theorem 3.3** (($p,\zeta$)-Sinkhorn bisimulation metrics). *Let $c_T \in [0, 1)$, $c_R \in [0, \infty)$, $p \geq 1$ and $\zeta \geq 0$. The mappings $\mathcal{F}, \mathcal{F}_\pi : \mathfrak{met}(\mathcal{S}) \to \mathfrak{met}(\mathcal{S})$ each have unique fixed-points:*

$$\mathcal{F}(d)(\boldsymbol{s}_i, \boldsymbol{s}_j) := \max_{\boldsymbol{a} \in \mathcal{A}} c_R |R(\boldsymbol{s}_i, \boldsymbol{a}) - R(\boldsymbol{s}_j, \boldsymbol{a})| + c_T W_p^\zeta(d)(\mathcal{P}(\boldsymbol{s}_i, \boldsymbol{a}), \mathcal{P}(\boldsymbol{s}_j, \boldsymbol{a})), \tag{10}$$

$$\mathcal{F}_\pi(d)(\boldsymbol{s}_i, \boldsymbol{s}_j) := c_R |R_\pi(\boldsymbol{s}_i) - R_\pi(\boldsymbol{s}_j)| + c_T W_p^\zeta(d)(\mathcal{P}_\pi(\boldsymbol{s}_i), \mathcal{P}_\pi(\boldsymbol{s}_j)). \tag{11}$$

*Whenever $c_T \geq \gamma$, (4) and (6) hold for all $p \geq 1$ and $\zeta \geq 0$ for fixed-points $d^\sim$ and $d_\pi^\sim$ respectively.*

The existence and uniqueness proof follows from Lemma 3.2 and the Banach fixed-point theorem. We note the following relationship between bisimulation metrics that use different values ($p, \zeta$).

*Remark* 3.4. Given metrics $d_1^\sim$ and $d_2^\sim$ defined via $(p_1, \zeta_1)$ and $(p_2, \zeta_2)$, $(p_1, \zeta_1) \preceq (p_2, \zeta_2) \Rightarrow d_1^\sim \leq d_2^\sim$.

That is, $\epsilon$-aggregation of $\mathcal{S}$ under $d_2^\sim$ is finer-grained than under $d_1^\sim$. Thus any speedups obtained via $p > 1$ or $\lambda > 0$ come at the expense of possibly less efficient discretizations (larger $|\widetilde{\mathcal{S}}|$), although one still enjoys the same VFA bounds given in (4) and (6) for the same $\epsilon$. Interestingly, we recover MICo (Castro et al., 2021) as a special case of ($p,\zeta$)-Sinkhorn bisimulation metrics; when $p = 1$ and $\zeta \to \infty$, the optimal transport plan $\omega^* \to \mu_1 \otimes \mu_2$ and the ($p,\zeta$)-Sinkhorn distance becomes the expected distance over $\mu_1 \otimes \mu_2$. This was precisely the distance used by MICo to replace the costly 1-Wasserstein distance.[6] Hence, the more general form (10-11) with $\zeta \geq 0$ establishes a continuum of metrics that spans bisimulation metrics and MICo at its two extremes. Similarly to Cuturi (2013), we provide theoretical results for the primal Sinkhorn distance $W_p^\zeta(d)$, but for empirical analysis in Sec. 4 we use the dual distance $W_p^\lambda(d)$ with a fixed $\lambda$ rather than optimize the dual variable $\lambda$ for a fixed $\zeta$. In particular, we investigate the quality of metrics with varying $p$ and $\lambda$, and how their differences may influence API algorithms that rely on said metrics for VFA.

---

[6]This distance measure is also known as the Łukaszyk-Karmowski distance (Łukaszyk, 2004; Castro et al., 2021).

## 3.2 Approximate Policy Iteration with $\pi$-bisimulation

Now that we can approximate bisimulation metrics more efficiently using Sinkhorn distances, we introduce a feasible API procedure with $\pi$-bisimulation metrics. Using (6), we will derive error bounds on optimality. We write $\text{GreedyImprovement}(V, \delta)$ to indicate an approximate greedy update with respect to a function $V \in \mathcal{B}(\mathcal{S})$, which yields a policy $\pi_g$ over $\mathcal{S}$ such that $\|T_{\pi_g} V - TV\|_\infty \leq \delta$.

**Theorem 3.5** (API with $\pi$-bisimulation). *Let $c_R = 1$, $c_T = \gamma$ and $\{\pi_k\}_{k \in \mathbb{N}}$ be a sequence of policies generated with the following updates per step, where $d_0 = \mathbf{0} \in \mathfrak{met}(\mathcal{S})$, and $\epsilon \geq 0$ and $n \in \mathbb{N}_+$ are algorithm parameters. Let $c_n = \gamma^n/(1 - \gamma)$, and consider for any $p \geq 1$ and $\zeta \geq 0$:*

$$\widehat{d}_{\pi_k} \leftarrow \mathcal{F}^{(n)}_{\pi_k}(d_0) \tag{12}$$

$$\widetilde{\mathcal{S}}, \Phi_k \leftarrow \text{HardAggregation}(\mathcal{S}, \widehat{d}_{\pi_k}, \epsilon) \tag{13}$$

$$\widetilde{V}^{\pi_k} \leftarrow \text{PolicyEvaluation}(\widetilde{\mathcal{S}}, \pi_k) \tag{14}$$

$$\pi_{k+1} \leftarrow \text{GreedyImprovement}(\widetilde{V}^{\pi_k}_{\Phi_k}, \delta), \tag{15}$$

*and $\widetilde{V}^{\pi_k}_{\Phi_k} \in \mathcal{B}(\mathcal{S})$ is the composition of $\widetilde{V}^{\pi_k}$ and $\Phi_k$. If the sequence $\{\pi_k\}_{k \in \mathbb{N}}$ converges to a policy $\overline{\pi}$, we have*

$$\|V^{\overline{\pi}} - V^*\|_\infty \leq \frac{\delta}{1 - \gamma} + \frac{2\gamma(2\epsilon + c_n)}{(1 - \gamma)^2}. \tag{16}$$

*Otherwise, it has the following limiting bound,*

$$\limsup_{k \to \infty} \|V^{\pi_k} - V^*\|_\infty \leq \frac{\delta}{(1 - \gamma)^2} + \frac{2\gamma(2\epsilon + c_n)}{(1 - \gamma)^3}. \tag{17}$$

Here HardAggregation yields a partitioning $\widetilde{\mathcal{S}}$ of $\mathcal{S}$ under $\widehat{d}_{\pi_k}$ with partition radius at most $\epsilon$ (see Appendix D.2 for pseudocode of an implementation for finite $\mathcal{S}$). As $\epsilon \to 0$, policy evaluation becomes increasingly accurate. The number of metric learning updates $n$ also provides a trade-off between run-time and policy performance guarantees. Larger $n$ implies more accurate approximations of $d^\sim_{\pi_k}$ due to (12). This translates to a better bound on the worst-case error on $V^*$ due to the $c_n$ term in (16-17). Note that metric learning errors, as well as errors due to stochastic approximation of $R_\pi$ or inexact environment dynamics can be absorbed in $\epsilon$; see Lemma A.9 in Appendix A for a decomposition of error terms.

For ease of analysis, this procedure naively learns the approximate metric $\widehat{d}_{\pi_k}$ from scratch after each policy update (see (12)). In practice, we may wish to warm-start metric learning with updates $\widehat{d}_{\pi_k} \leftarrow \mathcal{F}^{(n)}_{\pi_k}(\widehat{d}_{\pi_{k-1}})$ to approximate $d^\sim_{\pi_k}$ in fewer iterations $n$, or we may be learning a parametrized metric via gradient descent as in the DBC algorithm (Zhang et al., 2021). To understand the tradeoff of such metric updates, we next derive a bound on how much the $\pi$-bisimulation metric changes when the underlying policy is changed.

**Lemma 3.6** (Comparing $\pi$-bisimulation metrics of different policies). *Let $\pi, \pi'$ be a pair of policies and $d^\sim_\pi, d^\sim_{\pi'}$ corresponding $\pi$-bisimulation metrics given by $p \in [1, \infty)$ and $\lambda = 0$. The difference between $d^\sim_\pi$ and $d^\sim_{\pi'}$ is bounded by $D^\infty_{\text{TV}}(\pi, \pi') = \sup_{\boldsymbol{s} \in \mathcal{S}} D_{\text{TV}}(\pi(\boldsymbol{s}), \pi'(\boldsymbol{s}))$, the worst-case total variation distance of $\pi$ and $\pi'$:*

$$\|d^\sim_\pi - d^\sim_{\pi'}\|_\infty \leq \frac{2c_R}{(1 - c_T)^2} D^\infty_{\text{TV}}(\pi, \pi')^{\frac{1}{p}}. \tag{18}$$

Here, (18) guarantees that small policy updates lead to small changes in the $\pi$-bisimulation metric. Thus, we conjecture that restricting the policy update size should keep the metric learning objective stable and result in a better performance guarantee when warm-starting is used for faster metric learning. Indeed, inspired by Scherrer (2014), we write an API($\alpha$) procedure, which constrains policy updates such that $D^\infty_{\text{TV}}(\pi_{k+1}, \pi_k) \leq \alpha, \forall k \in \mathbb{N}$. Given such updates, we are guaranteed to have $\|d^\sim_{\pi_k} - d^\sim_{\pi_{k-1}}\|_\infty \leq 2c_R \alpha^{\frac{1}{p}}/(1 - c_T)^2$ by (18), which can be leveraged to ensure that warm-starting metric learning updates provides a better asymptotic bound than (17) under some conditions. Before that, we pause for another lemma, which generalizes Propositions 2.4.3 and 2.4.4 of Bertsekas (2018a).

**Lemma 3.7** (Generalized API($\alpha$) bounds). *Let $V \in \mathcal{B}(\mathcal{S})$ and policies $\pi, \pi', \pi_g$ satisfy the following:*

$$\|V^\pi - V\|_\infty \leq \delta_{\text{PE}}$$
$$\|T_{\pi_g} V - TV\|_\infty \leq \delta_{\text{GI}}$$
$$\pi' = \alpha \pi_g + (1-\alpha)\pi,$$

*where $\alpha \in [0,1]$. Then,*

$$\|V^{\pi'} - V^*\|_\infty \leq (1 - \alpha + \alpha\gamma)\|V^\pi - V^*\|_\infty + \alpha \frac{\delta_{\text{GI}} + 2\gamma\delta_{\text{PE}}}{1-\gamma}. \tag{19}$$

*Next, consider an API($\alpha$) algorithm that generates a sequence of policies $\{\pi_k\}_{k\in\mathbb{N}}$ via functions $\{V_k\}_{k\in\mathbb{N}}$ with policy evaluation error $\|V^{\pi_k} - V_k\|_\infty \leq \delta_{\text{PE},k}$, approximate greedy updates with error $\|T_{\pi_{g,k}} V_k - TV_k\|_\infty \leq \delta_{\text{GI,k}}$ and policy updates $\pi_{k+1} \leftarrow \alpha\pi_{g,k} + (1-\alpha)\pi_k$. For any $\alpha \in (0,1]$,*

$$\limsup_{k\to\infty}\|V^{\pi_k} - V^*\|_\infty \leq \frac{\limsup_{k\to\infty} \delta_{\text{GI},k} + 2\gamma\delta_{\text{PE},k}}{(1-\gamma)^2}. \tag{20}$$

For $\alpha = 1$, (19) recovers Proposition 2.4.4 of Bertsekas (2018a) as a special case. (20) follows by setting $\pi' = \pi_{k+1}$, $\pi = \pi_k$ in (19) and taking a limit superior on both sides; it proves that the same asymptotic bound as Proposition 2.4.3 of Bertsekas (2018a) holds for API($\alpha$) with arbitrary $\alpha \in (0,1]$. Furthermore, it makes explicit that the oscillation amplitude due to PE and GI errors is modulated by $\alpha$.

**Theorem 3.8** (API($\alpha$) with $\pi$-bisimulation). *Under the same conventions as Thm. 3.5, let $n > \log(\frac{1-\gamma}{1+\gamma})/\log(\gamma)$ and $\bar{c}_n = (1+\gamma)c_n$. Given $\alpha = \left(\bar{\alpha}(1 - \bar{c}_n)(1-\gamma)/2\right)^p$ for some $\bar{\alpha} \in (0,1]$, $\lambda = 0$ and any $p \in [1,\infty)$:*

$$\hat{d}_{\pi_k} \leftarrow \mathcal{F}^{(n)}_{\pi_k}(\hat{d}_{\pi_{k-1}}) \tag{21}$$
$$\widetilde{\mathcal{S}}, \Phi_k \leftarrow \text{HardAggregation}(\mathcal{S}, \hat{d}_{\pi_k}, \epsilon) \tag{22}$$
$$\widetilde{V}^{\pi_k} \leftarrow \text{PolicyEvaluation}(\widetilde{\mathcal{S}}, \pi_k) \tag{23}$$
$$\pi_g \leftarrow \text{GreedyImprovement}(\widetilde{V}^{\pi_k}_{\Phi_k}, \delta) \tag{24}$$
$$\pi_{k+1} \leftarrow (1-\alpha)\pi_k + \alpha\pi_g. \tag{25}$$

*The sequence $\{\pi_k\}_{k\in\mathbb{N}}$ has the following limiting bound,*

$$\limsup_{k\to\infty}\|V^{\pi_k} - V^*\|_\infty \leq \frac{\delta}{(1-\gamma)^2} + \frac{2\gamma(2\epsilon + \bar{\alpha}c_n)}{(1-\gamma)^3}. \tag{26}$$

As expected, by exploiting the induced stability of the sequence $\{\hat{d}_{\pi_k}\}_{k\in\mathbb{N}}$ via a small $\alpha$, we obtain a better asymptotic bound for the same $n$ as compared to Thm. 3.5 (since $\bar{\alpha} \leq 1$). The trade-off here is that setting $\alpha$ too small can slow down policy improvement and require more policy updates to attain the asymptotic bound. In particular, the number of steps $k$ necessary to attain a fixed worst-case error bound scales as $1/\alpha$ due to the contraction rate $1 - \alpha + \alpha\gamma$ seen in (19). Note that $\alpha = 0$ would amount to no policy update and is therefore ruled out by assumption: $\bar{\alpha} \in (0,1]$. We omit the convergence case here, although it may be possible to obtain a bound that scales similarly to (16) by following Proposition 2.4.5 of Bertsekas (2018a).[7]

The bounds here are expressed in terms of the sup-norm for simplicity and represent the worst case. However stronger bounds in terms of $L_p$ norms can be derived following prior work, e.g., Munos (2003); Farahmand et al. (2010). Indeed, as noted by Bertsekas (2011), the limiting bounds in (17) and (26) can be conservative and quickly attained in practice. Nevertheless, they provide insight about co-learning bisimulation metrics and policies when one does not assume a continuously improving policy. In Section 4, we validate this point empirically with an implementation of the procedures described in Thms. 3.5 and 3.8.

---

[7]Bertsekas & Tsitsiklis (1996) note that convergence is quite uncommon unless approximation errors are extremely small.

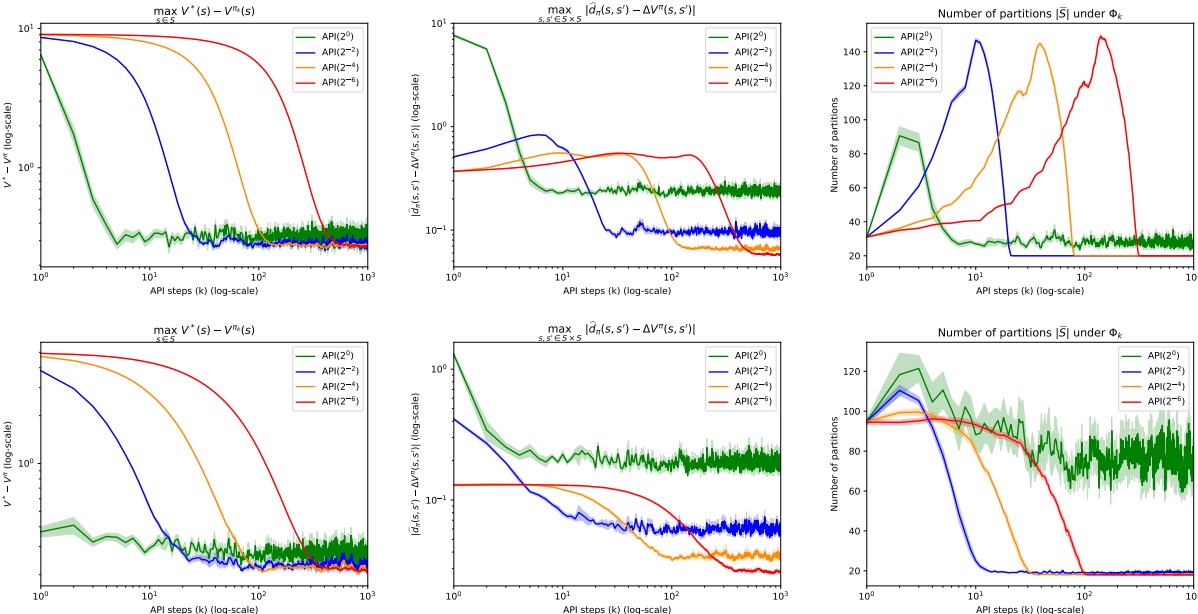

Figure 1: Ablation of $\alpha$ for the algorithm analyzed in Thm. 3.8 for two MDPs (top and bottom rows). **(Left)** While the algorithm reaches a similar performance range for all $\alpha$, we observe more oscillatory behavior for higher values of $\alpha$, which aligns with the ordering of asymptotic worst-case performance (limit superior) predicted by (26). The trade-off of reduced rate of improvement is also seen clearly. **(Middle)** The metric learned by lower $\alpha$ better approximates the value difference $\Delta V^\pi(\boldsymbol{s}, \boldsymbol{s}') = |V^\pi(\boldsymbol{s}) - V^\pi(\boldsymbol{s}')|$ both in terms of final error and variance in the limit. **(Right)** All $\alpha < 1$ shown here converge to a (near-)optimal partitioning of $\mathcal{S}$ with $|\widetilde{\mathcal{S}}| = m$, but $\alpha = 1$ is unable to find the optimal partitioning and has $\Phi_k$ oscillating for both MDPs.

### 3.3 Bridging Theory and Practice for Co-learning Policies and Bisimulation Metrics

Recall that Zhang et al. (2021) incorporated $\pi$-bisimulation metrics into the Soft actor-critic (SAC) algorithm (Haarnoja et al., 2018) via an auxiliary loss, which encourages the neural state representations $\phi(\boldsymbol{s})$ used by critic and (optionally) actor networks to respect an approximation $\widehat{d}_\pi$ of $d_{\widetilde{\pi}}$ computed with replay buffer samples and learned dynamics, i.e., $\|\phi(\boldsymbol{s}_i) - \phi(\boldsymbol{s}_j)\| \approx \widehat{d}_\pi(\boldsymbol{s}_i, \boldsymbol{s}_j)$. Such latent representations were shown to promote distraction invariance for continuous control tasks, which in turn improves sample efficiency and performance under heavy distraction over various representation learning baselines (Zhang et al., 2021), as well as the vanilla SAC algorithm (Kemertas & Aumentado-Armstrong, 2021). Similarly, MICo learned (in a value-based framework) a state encoder that respects another metric defined via $\mathbb{E}_{x_1 \sim \mu_1, x_2 \sim \mu_2}[d(x_1, x_2)]$ instead of $W_1(d)(\mu_1, \mu_2)$ for fast computation (Castro et al., 2021), which we connected to $(1, \zeta)$-Sinkhorn distances in Sec. 3.1 by taking a limit $\zeta \to \infty$. Here, we describe how the algorithm given in Thm. 3.8 maps onto actor-critic approaches used in applications.

The $n$-step fixed-point update given in (21) is feasible for sufficiently small finite $\mathcal{S}$, but not for large or continuous $\mathcal{S}$. Thus DBC computes the fixed-point target for a batch of states in a continuous setting, where $\widehat{d}_{\pi_k}(\boldsymbol{s}_i, \boldsymbol{s}_j) \coloneqq \|\phi_k(\boldsymbol{s}_i) - \phi_k(\boldsymbol{s}_j)\|$ and a sequence of $n$ fixed-point updates is replaced by a gradient update $\phi_{k+1} \leftarrow \phi_k + w\nabla J(\phi_k)$ with step-size $w$ for a loss,

$$J(\phi_k) = \frac{1}{2}\mathbb{E}\left[\left(\widehat{d}_{\pi_k}(\boldsymbol{s}_i, \boldsymbol{s}_j) - |R_\pi(\boldsymbol{s}_i) - R_\pi(\boldsymbol{s}_j)| - \gamma W_2(\widehat{d}_{\pi_k})(\widehat{\mathcal{P}}_\pi(\boldsymbol{s}_i), \widehat{\mathcal{P}}_\pi(\boldsymbol{s}_j))\right)^2\right],$$

where the expectation is estimated with replay buffer samples and $\widehat{\mathcal{P}}$ is a latent Gaussian model over $\phi$ space.

For VFA, we use a HardAggregation operation which explicitly computes $\widetilde{\mathcal{S}}$ and an aggregation function $\Phi$ as in (22). For finite MDPs, $\Phi$ is represented as an $|\widetilde{\mathcal{S}}| \times |\mathcal{S}|$ matrix with one-hot encoded columns, so that (23) is performed via value iteration over $\widetilde{R}_\pi$ and $\widetilde{\mathcal{P}}_\pi$ easily with matrix operations. The use of

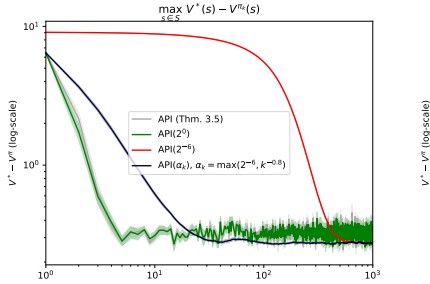 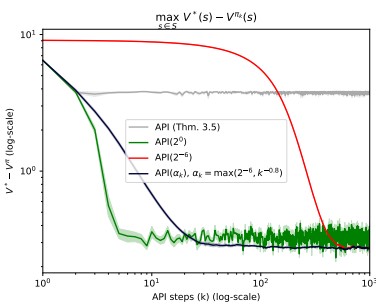 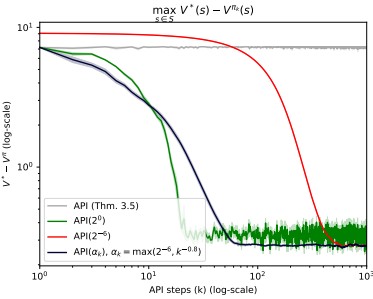

Figure 2: Comparing $\|V^* - V^{\pi_k}\|_\infty$ using API, API($\alpha$) and API($\alpha_k$) on the first MDP with $n \in [28, 7, 1]$ (**left-to-right**). See text.

(22-23) enables theoretical analysis via (6). On the other hand, practical approaches for continuous MDPs such as DBC learn a neural critic of the form $V_\theta(\phi(\boldsymbol{s}))$ with SGD, where $\phi(\boldsymbol{s})$ is trained to respect the bisimulation metric as described above. Hence, by construction we have $\widetilde{V}^\pi(\Phi(\boldsymbol{s}_i)) = \widetilde{V}^\pi(\Phi(\boldsymbol{s}_j))$ whenever $\Phi(\boldsymbol{s}_i) = \Phi(\boldsymbol{s}_j)$ such that $\widehat{d}_\pi(\boldsymbol{s}_i, \boldsymbol{s}_j) \leq 2\epsilon$. In contrast, in actor-critic based continuous control one softly promotes $V_\theta(\phi(\boldsymbol{s}_i)) \approx V_\theta(\phi(\boldsymbol{s}_j))$ for small $\widehat{d}_\pi = \|\phi(\boldsymbol{s}_i) - \phi(\boldsymbol{s}_j)\|$ via an architectural constraint on the critic.

Lastly, the operation GreedyImprovement($\widetilde{V}^{\pi_k}_{\Phi_k}, \delta$) in (24) can be viewed as a combination of an exact update $\widetilde{\pi}_g \leftarrow$ GreedyImprovement($\widetilde{V}^{\pi_k}, 0$) in the finite space $\widetilde{\mathcal{S}}$ followed by a lifting of this quantized policy to a policy $\pi_g$ over a possibly continuous $\mathcal{S}$ with some error $\delta \geq 0$. For practical continuous control, one noisily updates the parameters $\psi$ of an actor network $\pi_\psi$ with SGD to maximize values predicted by the bisimulation-constrained critic $V_\theta \circ \phi$. Computing an $\alpha$-mixture of policies as in (25) is trivial for finite MDPs as it amounts to a simple interpolation of probability vectors over the action space. However, iterative mixing of neural policies as in (25) over continuous MDPs is non-trivial. Hence, we focus on finite MDPs in the next section for a controlled empirical analysis of the theoretical results presented here.

## 4 Empirical Analysis

In this section, we conduct experiments to empirically investigate the practical implications of the theoretical results in Sec. 3. To this end, we consider a discounted problem involving $|\mathcal{S}|$ states and $m$ equivalence classes (ECs) denoted $B_i$ where $i \in [1, \ldots, m]$ and each EC contains the same number of states $|\mathcal{S}|/m$. At each step, an agent decides between two actions $\boldsymbol{a}_0$ and $\boldsymbol{a}_1$: taking $\boldsymbol{a}_0$ in $B_i$ transitions the agent to $B_{i+1}$, while $\boldsymbol{a}_1$ transitions the agent to $B_1$ (both with probability 1). Only when the agent takes $\boldsymbol{a}_0$ in $B_m$ does it obtain a reward of 1 and taking $\boldsymbol{a}_0$ keeps the agent in $B_m$ then. Hence, the agent is required to take $\boldsymbol{a}_0$ at least $m$ times consecutively to collect rewards after taking $\boldsymbol{a}_1$ once. When constructing $\mathcal{P}$, we sample uniformly from the $(|\mathcal{S}|/m-1)$-simplex to determine the transition probabilities to states in each EC. Therefore, each random seed generates an MDP with a different $\mathcal{P}$, but they all map to the same reduced MDP. We also consider a second MDP with dense rewards inspired by Example 6.4 of Bertsekas & Tsitsiklis (1996). This time, the agent stays in the same EC if it takes $\boldsymbol{a}_1$ rather than being transitioned to $B_1$. Once the agent reaches $B_m$, it stays there forever regardless of which action it takes. Rewards for when the agent takes $\boldsymbol{a}_0$ are defined recursively as $r_0 = e, r_{i+1} = \gamma r_i + e$, where we set $e = \frac{1-\gamma}{1-\gamma^m}$ so that $R \in [0, 1]$. We also consider a third class of MDPs in Appendix F and obtain similar results to those presented here.

In the experiments outlined here, we choose $|\mathcal{S}| = 200$, $m = 20$ and $\gamma = 0.9$. Unless otherwise stated, we use use $p = \lambda = 1$ and $n = \lceil \log(\frac{1-\gamma}{1+\gamma})/\log(\gamma) \rceil$ as in Thm. 3.8 ($n = 28$ for $\gamma = 0.9$). However, metric learning can be terminated early at a given step $k$ if $\|\mathcal{F}^{(i+1)}_{\pi_k}(d) - \mathcal{F}^{(i)}_{\pi_k}(d)\|_\infty \leq 10^{-3}$. To simulate noisy greedy updates, we perturb action probabilities of the ground-truth greedy policy with Gaussian noise and renormalize to form a distribution. A heuristic search of the noise variance ensures $\|T_{\pi_g} V - T V\|_\infty \in [0.05, 0.1]$ so that $\delta \leq 0.1$. In all cases, we initialize $\pi_0$ to be the maximum-entropy policy. Our source code will be open-sourced for reproducibility. We base our Sinkhorn-Knopp implementation on the Python Optimal Transport package (Flamary et al., 2021). All figures present results over 10 seeds with shaded areas showing standard error.

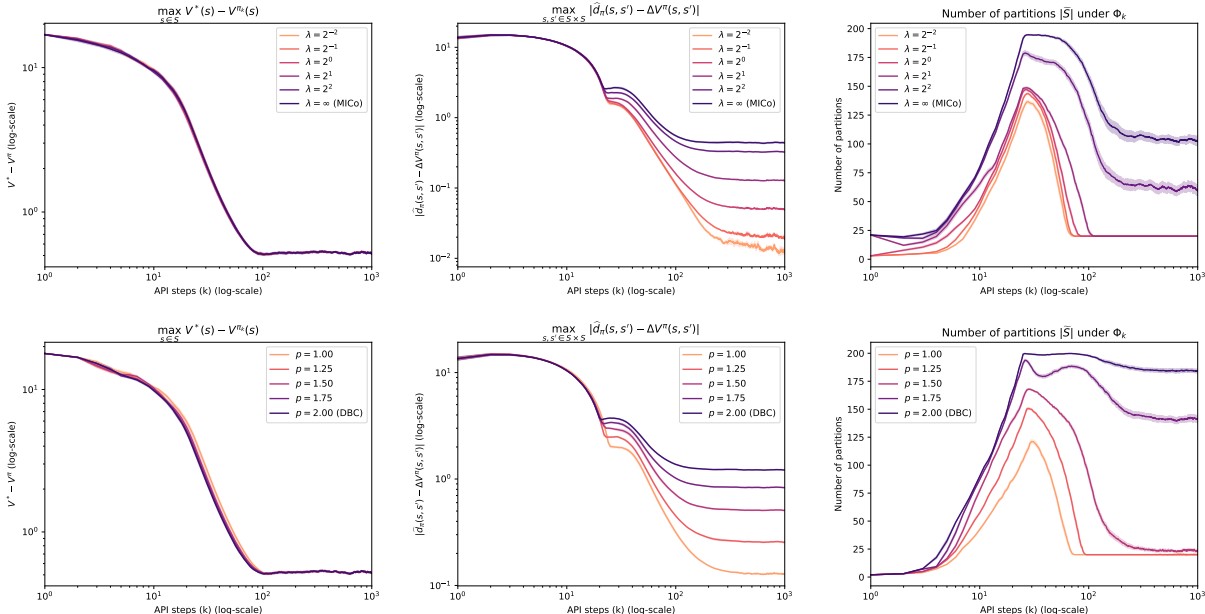

Figure 3: Similar to Fig. 1, but varying $\lambda$ for $p = 1$ and $\epsilon = 0.1$ **(top)**, and $p$ for $\lambda = 0.1$ and $\epsilon = 0.25$ **(bottom)** on the first MDP. See text.

**API, API($\alpha$) and API($\alpha_k$).** We begin our analysis by testing the algorithm in Thm. 3.8 with varying $\alpha$. The results shown in Fig. 1 confirm the key insights that emerged from the analysis in Sec. 3.2: (i) lower $\alpha$ makes progress more slowly, but is more stable and has better asymptotic worst-case performance, (ii) lower $\alpha$ better exploits warm-starting of metric learning to learn a higher quality metric, which in turn better approximates the value difference $\Delta V^\pi(\boldsymbol{s}, \boldsymbol{s}') = |V^\pi(\boldsymbol{s}) - V^\pi(\boldsymbol{s}')|$ (recall that the true metric $d_\pi^\sim$ satisfies $|V^\pi(\boldsymbol{s}) - V^\pi(\boldsymbol{s}')| \leq d_\pi^\sim(\boldsymbol{s}, \boldsymbol{s}')$ for $c_R = 1$, see also Appendix E), (iii) a higher quality metric makes for a better partitioning based on it so that all settings with $\alpha < 1$ recover the $m = 20$ ECs of the MDPs, while $\alpha = 1$ does not. Note that we use $\epsilon = 0.1$ for the first MDP and $\epsilon = 0.04$ for the second.

Next, we compare the algorithm in Thm. 3.8 to its naive version described in Thm. 3.5 for various settings of the number of fixed-point iterations $n$. We add a new variant called API($\alpha_k$) to the comparison based on the observation that we can leverage the high rate of improvement of a high $\alpha$ in early phases of learning and the improved stability of low $\alpha$ in the limit by scheduling $\alpha_k$ to decay gradually to a limiting $\alpha$. Indeed, given $\alpha_k \to \alpha$ the same asymptotic bound in (26) holds. In Fig. 2, we show that the naive algorithm behaves similarly to API(1.0) for $n$ large enough. However, for smaller $n$ the naive algorithm fails (cf. (16-17)), while the warm-start algorithm succeeds. Secondly, Fig. 2 shows that given a decay schedule such as $\alpha_k = \max(2^{-6}, k^{-0.8})$, we are able to obtain a better trade-off in terms of rate of improvement and stability than either of API(1.0) and API($2^{-6}$). We note that lower $n$ does not seem to compromise the warm-start algorithm stability, but only slightly slows down policy improvement in terms of time-steps $k$ for a good trade-off on wall-clock time (as fixed-point updates $\mathcal{F}_\pi$ comprise the main bottleneck on run-time). Lastly, we repeated the same experiment with $\gamma = 0.99$ and did not observe any qualitative differences.

$(p, \lambda)$**-Sinkhorn distances.** Here, our main goal is to investigate the results provided by Thm. 3.3 and Remark 3.4 in Sec. 3.1. As noted in Sec. 3.1, we use the dual Sinkhorn distance with a fixed $\lambda > 0$ for ease of computation and omit the optimization of the dual variable $\lambda$ for a fixed $\zeta$. However, we note the monotonic relationship between $\lambda$ and $\zeta$. In Fig. 3, we ablate $\lambda$ and $p$ to show how they influence learning for an API($\alpha$) algorithm. We use the API($\alpha_k$) variant described above with $\alpha_k = \max(0.01, k^{-0.8})$ as it strikes a better trade-off between rate of improvement and asymptotic stability than the fixed-$\alpha$ variant. Recall that all settings of $(p, \zeta)$ enjoy the same bound (6) as stated in Thm. 3.3, and consequently their usage in API also yields the same asymptotic performance in Thm. 3.8. In the left-most column of Fig. 3, we observe near-identical behavior in terms of optimality error $\|V^* - V^{\pi_k}\|_\infty$ not just asymptotically, but for all $k$ regardless of the choice of $(p, \lambda)$, i.e., learning dynamics are almost identical. However, the quality of the

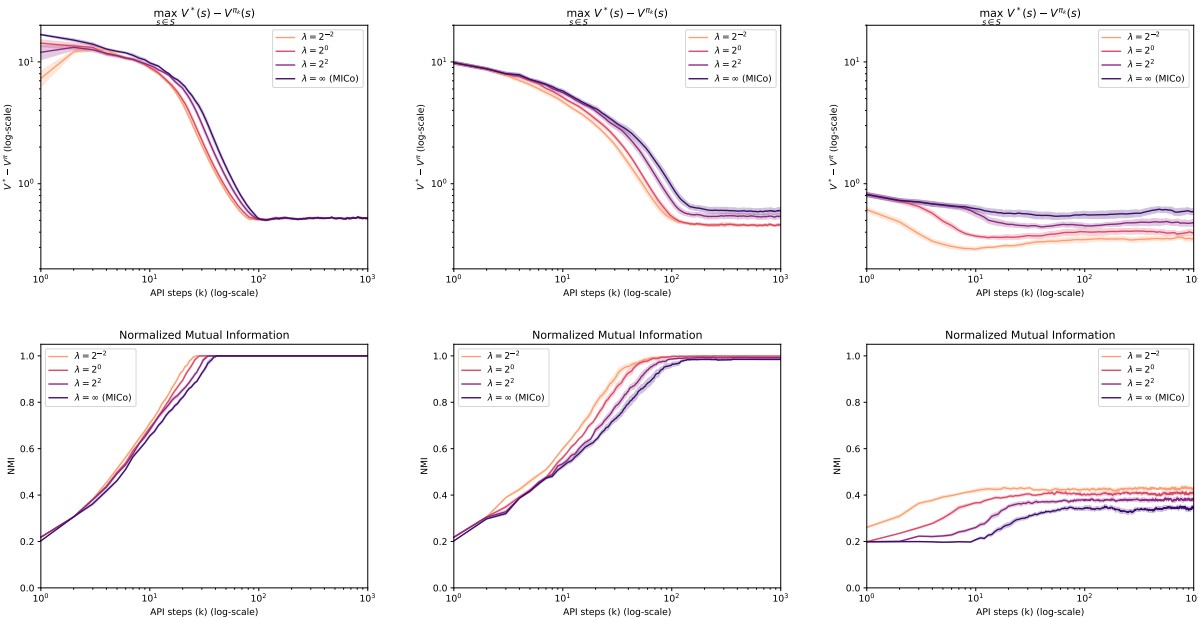

Figure 4: Varying $\lambda$ with unperturbed, lightly perturbed and heavily perturbed $\mathcal{P}$ **(left-to-right)** when $\epsilon$-aggregation is replaced by 30-medioids partitioning. **(Top)** Performance gap between different $\lambda$ settings widens with more stochastic $\mathcal{P}$. **(Bottom)** Quality of the learned metric as measured by normalized mutual information (NMI) of labels found by $m$-medioids with ground-truth EC labels of the unperturbed MDP.

metric decreases with increasing $p$ and $\lambda$ in terms of its ability to approximate the value difference between a pair of states (see Remark 3.4). As a result, given a fixed $\epsilon$ one obtains a less efficient discretization (finer-grained partitioning) of the state space. The output of bisimulation-based RL algorithms is not only a policy optimized to solve a certain task, but also an efficient state encoder (or aggregator) that ignores functionally-irrelevant aspects of the environment state; the experiments here show that while all of these metrics (when learned with the same number of fixed-point iterations) follow a nearly identical policy optimization path, the choice of $(p, \lambda)$ defines a trade-off between time and space complexity for hard-aggregation with $\Phi$, and possibly time complexity and encoder *quality* for representation learning with state encoders $\phi$.

**Aggregation on a space budget.** In Figs. 1 and 3, we observe that the algorithm often computes a fine-grained partitioning ($|\widetilde{\mathcal{S}}|$ close to $|\mathcal{S}| = 200$) early on given a fixed $\epsilon$. We inquire whether this costly phase is necessary to find the optimal policy in $\pi$-bisimulation-based API. Secondly, we also observed finer-grained partitioning with increasing $\lambda$. We thus inquire whether the metric learned by MICo ($\lambda \to \infty$) still captures the same geometric information about the state space but only on a different scale, or it actually loses information about state similarities due to a weaker upper bound on $\Delta V^\pi$. To investigate these questions, we test a variant of the algorithm; we replace $\epsilon$-aggregation as in (22) with partitioning around medioids (PAM) with a fixed number of 30 partitions (Kaufman & Rousseeuw, 1990). Thirdly, in Appendix C, we identify a potential shortcoming of the Sinkhorn distance: guided by information-theoretic intuition, we find that the Sinkhorn distance tends to compute a weaker upper bound on the Wasserstein distance when the distributions being compared have higher entropy. We thus consider MDPs with varying degrees of stochasticity in $\mathcal{P}$: in particular, we perturb each transition matrix $\mathcal{P}(\cdot, \boldsymbol{a})$ from the first MDP, which is deterministic, by mixing it with randomly sampled transition matrices (each row sampled uniformly from the $(|\mathcal{S}|-1)$-simplex) with mixture weights 0.05 and 0.5. Conclusions from Fig. 4 are threefold. (i) The algorithm makes similar progress towards solving the task given a partition budget $50\%$ above $m$, i.e., it does not strictly require too many partitions early on to make progress later. (ii) We confirm that metrics with smaller $\lambda$ yield a more informative partitioning, and observe correlation between the quality of the metric and performance. In other words, if representation capacity allocated to $\Phi$ and $\widetilde{V}_\Phi^\pi$ is constrained, lower $\lambda$ produces better results at the expense of some added computation time (more Sinkhorn-Knopp iterations). (iii) The Sinkhorn distance offers a better complexity-performance trade-off for more deterministic MDPs.

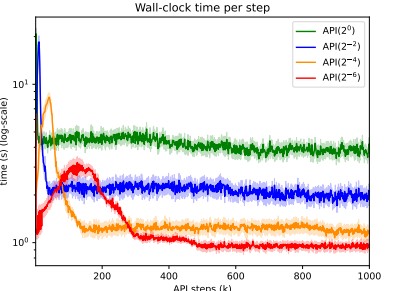 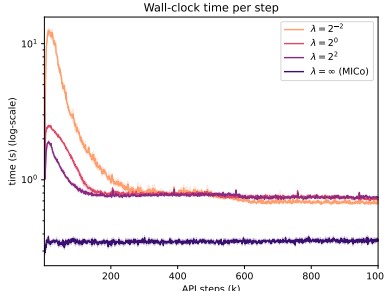

Figure 5: Better runtime with lower $\alpha$ (**Left**) and higher $\lambda$ with decaying $\alpha_k$ as before (**Right**) on an NVIDIA GeForce GTX 1080 GPU.

**Runtime measurements.** In Sec. 3.1, we discussed how one can warm-start the Sinkhorn-Knopp algorithm with Sinkhorn potentials of previous metrics at some memory cost (see Appendix D.1 for details). A second advantage involves the use of conservative policy updates. Intuitively, when $\pi_k$, $\mathcal{P}_{\pi_k}$ and the ground metric $\widehat{d}_{\pi_k}$ do not change much over time-steps $k$, the Sinkhorn potentials corresponding to $W_p^\lambda(\widehat{d}_{\pi_k})(\mathcal{P}_{\pi_k}(\boldsymbol{s}), \mathcal{P}_{\pi_k}(\boldsymbol{s}'))$ should not either. In light of Lemma 3.6, we suspect that with conservative policy updates, the initial guess provided to the Sinkhorn-Knopp algorithm might be closer to the true solution when conservative policy updates are employed, which may result in faster convergence. In Fig. 5, we measure the wall-clock time per iteration of the API($\alpha$) algorithm to validate this hypothesis. Secondly, we noted in Sec. 3.1 that greater $\lambda$ requires fewer steps for Sinkhorn-Knopp to converge; this is also demonstrated in Fig. 5. However, the cost of a small $\lambda$ is eventually amortized by the warm-start strategy as the metric converges (after $\approx 300$ steps). Note that for $\lambda \to \infty$, we have a closed-form for finite spaces (Cuturi, 2013):

$$\lim_{\lambda \to \infty} W_1^\lambda(d)(\mu_1, \mu_2) = \lim_{\zeta \to \infty} W_1^\zeta(d)(\mu_1, \mu_2) = \mu_1^T D \mu_2,$$

where $D$ is a distance matrix with $D_{ij} = d(x_i, x_j)$, and $\mu_1$ and $\mu_2$ are probability vectors. Hence, in this case one bypasses Sinkhorn-Knopp entirely. The computation of the metric $W_1^\lambda(d)(\mathcal{P}_\pi(\boldsymbol{s}), \mathcal{P}_\pi(\boldsymbol{s}'))$ for all state pairs can be easily parallelized for finite MDPs; all pairwise distances can be computed as $\mathcal{P}_\pi D \mathcal{P}_\pi^T$ so that the case $\lambda \to \infty$ can be taken as a best-case run-time benchmark for our implementation.

## 5 Conclusion

In this work, we analyzed the use of bisimulation metrics in approximate policy iteration to bridge the gap between theory and practice. We first generalized bisimulation metrics to $(p, \zeta)$-Sinkhorn distances where $p \geq 1$ and $\zeta \geq 0$. The $p$-Wasserstein generalization confirmed the theoretical results on VFA of Kemertas & Aumentado-Armstrong (2021) given in (6) and added theoretical justification to the use of 2-Wasserstein metrics for fast computation as in Zhang et al. (2021). Sinkhorn distances enabled GPU-based fast approximation of upper bounds on $p$-Wasserstein bisimulation metrics with better time complexity than prior work (Ferns et al., 2004), and established a theoretical formalism for a more general family of metrics encompassing standard bisimulation metrics and MICo (Castro et al., 2021). We further conducted a theoretical analysis of API procedures that use a bisimulation-based discretization of the state space for VFA. The analysis indicated that conservative updates may benefit such procedures since a rapidly changing policy makes for a rapidly changing metric learning objective. Indeed, we showed that conservative updates enable warm-starting of metric learning iterations with significantly better speedup-performance trade-offs. To validate our theoretical findings and investigate trade-offs, we implemented the theoretically-grounded API($\alpha$) algorithm, which mimics the actor-critic based applications of bisimulation-like metrics for continuous control (Zhang et al., 2021; Kemertas & Aumentado-Armstrong, 2021; Castro et al., 2021). We provided an ablation analysis of algorithm parameters, and also showed empirically that decaying policy update sizes may strike better trade-offs between asymptotic performance, stability and rate of improvement. Further, in our controlled setting we showed that metric learning speedups are gained by a trade-off in the quality of the learned metric and space complexity of corresponding state abstractions. Whenever VFA capacity (as

measured here by the number of allowed partitions) is sufficient, the policy performance for the task at hand remains unaffected by the use of weaker metrics; however, capacity limitations may result in performance degradation as was shown in Fig. 4. Furthermore, we presented evidence that the Sinkhorn distance may offer a better trade-off under more deterministic transitions. While we focused on theoretical analysis in this work and limited our empirical analysis to finite MDPs, future work may consider practical approaches to implementing bisimulation-based actor-critic algorithms with conservative policy updates, and sample-based approximation of the Sinkhorn distance between continuous distributions for continuous control.

**Acknowledgements.** We acknowledge the support of the Natural Sciences and Engineering Research Council of Canada (NSERC).

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

## A  Proofs and Additional Results

**Lemma 3.2** (A $(p, \zeta)$-Sinkhorn distance bound)**.** *Given metrics $d$ and $d'$, for all $p \geq 1$ and $\zeta \geq 0$,*

$$\left| W_p^\zeta(d)(\mu_1, \mu_2) - W_p^\zeta(d')(\mu_1, \mu_2) \right| \leq \|d - d'\|_\infty. \tag{9}$$

*Proof.* Let $\Omega(\zeta) = \{\omega \in \Omega \mid D_{\mathrm{KL}}(\omega \,||\, \mu_1 \otimes \mu_2) \leq \zeta^{-1}\}$.

$$
\begin{aligned}
& W_p^\zeta(d)(\mu_1, \mu_2) - W_p^\zeta(d')(\mu_1, \mu_2) \\
&= \min_{\omega \in \Omega(\zeta)} \|d\|_{p,\omega} - W_p^\zeta(d')(\mu_1, \mu_2) \\
&= \min_{\omega \in \Omega(\zeta)} \|d - d' + d'\|_{p,\omega} - W_p^\zeta(d')(\mu_1, \mu_2) \\
&\leq \min_{\omega \in \Omega(\zeta)} \left( \|d - d'\|_{p,\omega} + \|d'\|_{p,\omega} \right) - W_p^\zeta(d')(\mu_1, \mu_2) && \text{(since triangle inequality holds for all } \omega \in \Omega(\zeta)) \\
&\leq \min_{\omega \in \Omega(\zeta)} \left( \|d - d'\|_\infty + \|d'\|_{p,\omega} \right) - W_p^\zeta(d')(\mu_1, \mu_2) && \text{(since } \|\cdot\|_{p,\omega} \leq \|\cdot\|_\infty) \\
&= \|d - d'\|_\infty + \min_{\omega \in \Omega(\zeta)} \|d'\|_{p,\omega} - W_p^\zeta(d')(\mu_1, \mu_2) \\
&= \|d - d'\|_\infty. && \blacksquare
\end{aligned}
$$

**Lemma A.1** (Triangle inequality for $W_p^\zeta(d)$)**.** *Consider three probability measures $\mu_1, \mu_2, \mu_3 \in \mathcal{P}_p(\mathcal{X})$.*

$$W_p^\zeta(d)(\mu_1, \mu_3) \leq W_p^\zeta(d)(\mu_1, \mu_2) + W_p^\zeta(d)(\mu_2, \mu_3). \tag{27}$$

*Proof.* For this proof, we use the Gluing Lemma (see Chapter 1 of Villani (2008)) and Lemma 1 of Cuturi (2013). Note that triangle inequality readily holds for $W_p(d)$ (Villani, 2008) and $W_1^\zeta(d)$ (Cuturi, 2013).

Let random variables $(X_1, X_2)$ be the optimal coupling of $(\mu_1, \mu_2)$ and $(Z_2, Z_3)$ the optimal coupling of $(\mu_2, \mu_3)$. By the Gluing Lemma, there exist random variables $(X_1', X_2', X_3')$ such that $\omega_{12}^* = \mathrm{law}(X_1', X_2') = \mathrm{law}(X_1, X_2)$ and $\omega_{23}^* = \mathrm{law}(X_2', X_3') = \mathrm{law}(Z_2, Z_3)$, where $\omega_{12}^*, \omega_{23}^* \in \Omega(\zeta)$. Furthermore, by Lemma 1 of Cuturi (2013), $(X_1', X_3')$ is a coupling of $(\mu_1, \mu_3)$ such that $\omega_{13} = \mathrm{law}(X_1', X_3') \in \Omega(\zeta)$. That is, $\omega_{13}$ also satisfies the entropic constraint given by $\zeta$ and is therefore feasible.

$$
\begin{aligned}
W_p^\zeta(d)(\mu_1, \mu_3) &\leq \left( \mathbb{E}[d(X_1', X_3')^p] \right)^{\frac{1}{p}} && \text{(since the coupling } (X_1', X_3') \text{ is not necessarily optimal for } (\mu_1, \mu_3)) \\
&\leq \left( \mathbb{E}\left[ \left( d(X_1', X_2') + d(X_2', X_3') \right)^p \right] \right)^{\frac{1}{p}} && \text{(by triangle inequality since } d \text{ is a pseudo-metric)} \\
&\leq \left( \mathbb{E}[d(X_1', X_2')^p] \right)^{\frac{1}{p}} + \left( \mathbb{E}[d(X_2', X_3')^p] \right)^{\frac{1}{p}} && \text{(by the Minkowski inequality)} \\
&= W_p^\zeta(d)(\mu_1, \mu_2) + W_p^\zeta(d)(\mu_2, \mu_3),
\end{aligned}
$$

where the equality in the final step holds since the couplings $(X_1', X_2')$ and $(X_2', X_3')$ are optimal with respect to $(\mu_1, \mu_2)$ and $(\mu_2, \mu_3)$ respectively by construction. $\blacksquare$

**Corollary A.2** (Approximate metrics and approximate distributions)**.** *Consider distributions $\mu_1, \overline{\mu}_1, \mu_2, \overline{\mu}_2 \in \mathcal{P}_p(\mathcal{X})$ and a pair of metrics $d, d'$. Due to Lemma 3.2 and Lemma A.1,*

$$|W_p^\zeta(d)(\mu_1, \mu_2) - W_p^\zeta(d')(\overline{\mu}_1, \overline{\mu}_2)| \leq W_p^\zeta(d)(\mu_1, \overline{\mu}_1) + W_p^\zeta(d)(\mu_2, \overline{\mu}_2) + \|d - d'\|_\infty. \tag{28}$$

*Proof.*

$$|W_p^\zeta(d)(\mu_1, \mu_2) - W_p^\zeta(d')(\overline{\mu}_1, \overline{\mu}_2)|$$

$$= \left| W_p^\zeta(d)(\mu_1, \mu_2) - W_p^\zeta(d)(\overline{\mu}_1, \overline{\mu}_2) + W_p^\zeta(d)(\overline{\mu}_1, \overline{\mu}_2) - W_p^\zeta(d')(\overline{\mu}_1, \overline{\mu}_2) \right|$$

$$\leq \left| W_p^\zeta(d)(\mu_1, \mu_2) - W_p^\zeta(d)(\overline{\mu}_1, \overline{\mu}_2) \right| + \left| W_p^\zeta(d)(\overline{\mu}_1, \overline{\mu}_2) - W_p^\zeta(d')(\overline{\mu}_1, \overline{\mu}_2) \right| \qquad \text{(triangle inequality)}$$

$$\leq \left| W_p^\zeta(d)(\mu_1, \mu_2) - W_p^\zeta(d)(\overline{\mu}_1, \overline{\mu}_2) \right| + \|d - d'\|_\infty \qquad \text{(by Lemma 3.2)}$$

$$\leq \left| W_p^\zeta(d)(\mu_1, \overline{\mu}_1) + W_p^\zeta(d)(\overline{\mu}_1, \overline{\mu}_2) + W_p^\zeta(d)(\overline{\mu}_2, \mu_2) - W_p^\zeta(d)(\overline{\mu}_1, \overline{\mu}_2) \right| + \|d - d'\|_\infty \qquad \text{(by Lemma A.1)}$$

$$= W_p^\zeta(d)(\mu_1, \overline{\mu}_1) + W_p^\zeta(d)(\mu_2, \overline{\mu}_2) + \|d - d'\|_\infty. \qquad \blacksquare$$

**Theorem 3.3** $((p, \zeta)$-Sinkhorn bisimulation metrics)**.** *Let* $c_T \in [0, 1)$, $c_R \in [0, \infty)$, $p \geq 1$ *and* $\zeta \geq 0$. *The mappings* $\mathcal{F}, \mathcal{F}_\pi : \mathfrak{met}(\mathcal{S}) \to \mathfrak{met}(\mathcal{S})$ *each have unique fixed-points:*

$$\mathcal{F}(d)(\boldsymbol{s}_i, \boldsymbol{s}_j) \coloneqq \max_{\boldsymbol{a} \in \mathcal{A}} c_R |R(\boldsymbol{s}_i, \boldsymbol{a}) - R(\boldsymbol{s}_j, \boldsymbol{a})| + c_T W_p^\zeta(d)(\mathcal{P}(\boldsymbol{s}_i, \boldsymbol{a}), \mathcal{P}(\boldsymbol{s}_j, \boldsymbol{a})), \tag{10}$$

$$\mathcal{F}_\pi(d)(\boldsymbol{s}_i, \boldsymbol{s}_j) \coloneqq c_R |R_\pi(\boldsymbol{s}_i) - R_\pi(\boldsymbol{s}_j)| + c_T W_p^\zeta(d)(\mathcal{P}_\pi(\boldsymbol{s}_i), \mathcal{P}_\pi(\boldsymbol{s}_j)). \tag{11}$$

*Whenever* $c_T \geq \gamma$, *(4) and (6) hold for all* $p \geq 1$ *and* $\zeta \geq 0$ *for fixed-points* $d^\sim$ *and* $d_\pi^\sim$ *respectively.*

*Proof.* Similarly to the proofs of Thm. 3.12 of Ferns et al. (2011) and Remark 1 of Kemertas & Aumentado-Armstrong (2021), it suffices to show that above fixed-point updates are contraction mappings. Then, we invoke the Banach fixed-point theorem to show the existence of a unique metric. Intuitively, applying the operator $\mathcal{F}$ on different metrics $d, d' \in \mathfrak{met}(\mathcal{S})$ should bring the metrics closer under the supremum (uniform) norm, such that $\lim_{n \to \infty} \mathcal{F}^{(n)}(d) = \lim_{n \to \infty} \mathcal{F}^{(n)}(d')$, i.e., they converge to the same unique metric. Here, compactness of $\mathcal{S}$ implies that $\mathfrak{met}(\mathcal{S})$ is complete such that the Banach fixed-point theorem can be applied (Ferns et al., 2011). Due to Lemma 3.2,

$$\mathcal{F}_\pi(d)(\boldsymbol{s}_i, \boldsymbol{s}_j) - \mathcal{F}_\pi(d')(\boldsymbol{s}_i, \boldsymbol{s}_j) = c_T \left( W_p(d)(\mathcal{P}_\pi(\boldsymbol{s}_i), \mathcal{P}_\pi(\boldsymbol{s}_j)) - W_p(d')(\mathcal{P}_\pi(\boldsymbol{s}_i), \mathcal{P}_\pi(\boldsymbol{s}_j)) \right)$$
$$\leq c_T \|d - d'\|_\infty, \ \forall (\boldsymbol{s}_i, \boldsymbol{s}_j) \in \mathcal{S} \times \mathcal{S}.$$

Taking a supremum on the LHS over $\mathcal{S} \times \mathcal{S}$, we obtain $\|\mathcal{F}_\pi(d) - \mathcal{F}_\pi(d')\|_\infty \leq c_T \|d - d'\|_\infty$, i.e., $\mathcal{F}_\pi$ is a $c_T$-contraction with respect to the sup-norm. Then, $\mathcal{F}_\pi$ has a unique fixed-point due to the Banach-fixed point theorem. The same result readily applies to $\mathcal{F}$ as well.

We now prove (6) for $\mathcal{F}_\pi$. First, note that by definition of the primal Sinkhorn distance in (7), we have for any $p, d, \mu_1, \mu_2$

$$\zeta_1 \leq \zeta_2 \Rightarrow \Omega(\zeta_2) \subseteq \Omega(\zeta_1) \Rightarrow W_p^{\zeta_1}(d)(\mu_1, \mu_2) \leq W_p^{\zeta_2}(d)(\mu_1, \mu_2).$$

Then, given fixed-point metrics $d_1$ and $d_2$ given by $(p, \zeta_1)$ and $(p, \zeta_2)$, we have $\zeta_1 \leq \zeta_2 \Rightarrow d_1 \leq d_2$. More generally, we have $(p_1, \zeta_1) \preceq (p_2, \zeta_2) \Rightarrow d_1 \leq d_2$ as noted in Remark 3.4 since $p_1 \leq p_2 \Rightarrow W_{p_1}(d)(\mu_1, \mu_2) \leq W_{p_2}(d)(\mu_1, \mu_2)$ (Villani, 2008). But from Castro (2020) and Kemertas & Aumentado-Armstrong (2021) we have for any $\pi$, $c_R |V^\pi(\boldsymbol{s}) - V^\pi(\boldsymbol{s}')| \leq d_\pi^\sim(\boldsymbol{s}, \boldsymbol{s}')$ for the special case when $p = 1$ and $\zeta$ is sufficiently small so that the constraint is satisfied for all $\omega \in \Omega$ and $W_p^\zeta(d)(\mu_1, \mu_2) = W_p(d)(\mu_1, \mu_2)$. Then, we have $(p_1, \zeta_1) \preceq (p_2, \zeta_2) \Rightarrow c_R |V^\pi(\boldsymbol{s}) - V^\pi(\boldsymbol{s}')| \leq d_1 \leq d_2$ so that all metrics defined by $(p, \zeta)$ provide an upper bound on $c_R |V^\pi(\boldsymbol{s}) - V^\pi(\boldsymbol{s}')|$. The result (6) follows immediately from Lipschitz continuity of $V_\pi$ with respect to $d_\pi^\sim$ (see proof of Lemma 8 by Kemertas & Aumentado-Armstrong (2021)), which is true of all metrics given by $(p, \zeta)$; hence (6) holds for all $(p, \zeta)$. The same logic applies to $\mathcal{F}$ and $V^*$ due to Ferns et al. (2011), so we skip the proof of (4) for brevity. $\blacksquare$

**Definition A.3** (Total variation distance (Gibbs & Su, 2002))**.** The total variation distance between a pair of distributions $\mu_1, \mu_2$ over a measurable space $\mathcal{X}$ is given by the following:

$$D_{\text{TV}}(\mu_1, \mu_2) = \sup_{A \subset \mathcal{X}} |\mu_1(A) - \mu_2(A)| \tag{29}$$

$$= \frac{1}{2} \max_{f \in \mathcal{F}} \left| \int_\mathcal{X} f(\boldsymbol{x})(\mu_1(\boldsymbol{x}) - \mu_2(\boldsymbol{x})) d\boldsymbol{x} \right| \tag{30}$$

where $\mathcal{F} = \{f : \mathcal{X} \to \mathbb{R} \mid \|f\|_\infty \leq 1\}$. For discrete distributions, $D_{\mathrm{TV}}(\mu_1, \mu_2) = \frac{1}{2}\|\mu_1 - \mu_2\|_1$.

**Corollary A.4** (Total variation distance to a mixture distribution). *Given distributions $\mu_1, \mu_2$ over a measurable space $\mathcal{X}$ and a scalar $\alpha \in [0,1]$:*

$$D_{\mathrm{TV}}(\mu_1, (1-\alpha)\mu_1 + \alpha\mu_2) = \alpha D_{\mathrm{TV}}(\mu_1, \mu_2) \tag{31}$$

*Proof.* Follows immediately from (30). ∎

**Lemma A.5** (Total variation distance between policies vs. transition distributions). *Consider a pair of policies $\pi, \pi'$ and the policy-dependent transition distributions $\mathcal{P}_\pi(\boldsymbol{s}), \mathcal{P}_{\pi'}(\boldsymbol{s})$,*

$$D_{\mathrm{TV}}(\mathcal{P}_\pi(\boldsymbol{s}), \mathcal{P}_{\pi'}(\boldsymbol{s})) \leq D_{\mathrm{TV}}(\pi(\boldsymbol{s}), \pi'(\boldsymbol{s})). \tag{32}$$

*Proof.* Let $\mathcal{F} = \{f : \mathcal{S} \to \mathbb{R} \mid \|f\|_\infty \leq 1\}$ and $\mathcal{G} = \{g : \mathcal{A} \to \mathbb{R} \mid \|g\|_\infty \leq 1\}$. Using Definition A.3 of the total variation distance,

$$
\begin{aligned}
&D_{\mathrm{TV}}(\mathcal{P}_\pi(\boldsymbol{s}), \mathcal{P}_{\pi'}(\boldsymbol{s})) \\
&= \frac{1}{2} \max_{f \in \mathcal{F}} \left| \int_{\mathcal{S}} f(\boldsymbol{s}')(\mathcal{P}_\pi(\boldsymbol{s}'|\boldsymbol{s}) - \mathcal{P}_\pi(\boldsymbol{s}'|\boldsymbol{s}))d\boldsymbol{s}' \right| \\
&= \frac{1}{2} \max_{f \in \mathcal{F}} \left| \int_{\mathcal{S}} f(\boldsymbol{s}') \int_{\mathcal{A}} \mathcal{P}(\boldsymbol{s}'|\boldsymbol{s}, \boldsymbol{a}) \Big( \pi(\boldsymbol{a}|\boldsymbol{s}) - \pi'(\boldsymbol{a}|\boldsymbol{s}) \Big) d\boldsymbol{a} d\boldsymbol{s}' \right| \\
&= \frac{1}{2} \max_{f \in \mathcal{F}} \left| \int_{\mathcal{A}} \Big( \pi(\boldsymbol{a}|\boldsymbol{s}) - \pi'(\boldsymbol{a}|\boldsymbol{s}) \Big) \int_{\mathcal{S}} f(\boldsymbol{s}')\mathcal{P}(\boldsymbol{s}'|\boldsymbol{s}, \boldsymbol{a})d\boldsymbol{s}' d\boldsymbol{a} \right| && \text{(by Fubini's theorem)} \\
&= \frac{1}{2} \left| \int_{\mathcal{A}} \Big( \pi(\boldsymbol{a}|\boldsymbol{s}) - \pi'(\boldsymbol{a}|\boldsymbol{s}) \Big) \underbrace{\int_{\mathcal{S}} f^*(\boldsymbol{s}')\mathcal{P}(\boldsymbol{s}'|\boldsymbol{s}, \boldsymbol{a})d\boldsymbol{s}'}_{=\tilde{f}(\boldsymbol{s}, \boldsymbol{a})} d\boldsymbol{a} \right| \\
&\leq \frac{1}{2} \left| \int_{\mathcal{A}} \Big( \pi(\boldsymbol{a}|\boldsymbol{s}) - \pi'(\boldsymbol{a}|\boldsymbol{s}) \Big) \underbrace{\max_{\boldsymbol{s} \in \mathcal{S}} \tilde{f}(\boldsymbol{s}, \boldsymbol{a})}_{=\tilde{g}(\boldsymbol{a})} d\boldsymbol{a} \right| \\
&\leq \frac{1}{2} \max_{g \in \mathcal{G}} \left| \int_{\mathcal{A}} \Big( \pi(\boldsymbol{a}|\boldsymbol{s}) - \pi'(\boldsymbol{a}|\boldsymbol{s}) \Big) g(\boldsymbol{a})d\boldsymbol{a} \right| \\
&= D_{\mathrm{TV}}(\pi(\boldsymbol{s}), \pi'(\boldsymbol{s})),
\end{aligned}
$$

where we used the fact that $\|\tilde{f}\|_\infty \leq 1$ and $\|\tilde{g}\|_\infty \leq 1$. ∎

**Lemma A.6** (Total variation distance between policies vs. immediate reward difference). *Consider a pair of policies $\pi, \pi'$ and the policy-dependent reward functions $R_\pi, R_{\pi'}$,*

$$|R_\pi(\boldsymbol{s}) - R_{\pi'}(\boldsymbol{s})| \leq D_{\mathrm{TV}}(\pi(\boldsymbol{s}), \pi'(\boldsymbol{s})). \tag{33}$$

*Proof.*

$$
\begin{aligned}
|R_\pi(\boldsymbol{s}) - R_{\pi'}(\boldsymbol{s})| &= \left| \int_{\mathcal{A}} (\pi(\boldsymbol{a}|\boldsymbol{s}) - \pi'(\boldsymbol{a}|\boldsymbol{s})) R(\boldsymbol{s}, \boldsymbol{a})d\boldsymbol{a} \right| \\
&= \frac{1}{2} \left| \int_{\mathcal{A}} (\pi(\boldsymbol{a}|\boldsymbol{s}) - \pi'(\boldsymbol{a}|\boldsymbol{s})) \, 2R(\boldsymbol{s}, \boldsymbol{a})d\boldsymbol{a} \right| \\
&= \frac{1}{2} \left| \int_{\mathcal{A}} (\pi(\boldsymbol{a}|\boldsymbol{s}) - \pi'(\boldsymbol{a}|\boldsymbol{s})) \, (2R(\boldsymbol{s}, \boldsymbol{a}) - 1)d\boldsymbol{a} \right| && \text{(since } \pi \text{ and } \pi' \text{ both integrate to 1)} \\
&\leq D_{\mathrm{TV}}\Big( \pi(\boldsymbol{s}), \pi'(\boldsymbol{s}) \Big),
\end{aligned}
$$

where the last inequality is due to Definition A.3 and the fact that $\|2R - 1\|_\infty \leq 1$ given $R \in [0,1]$. ∎

**Lemma A.7** (Wasserstein distance of transition distributions under different policies)**.** *Given a pair of policies* $\pi, \pi'$ *and the policy-dependent transition distributions* $\mathcal{P}_\pi(\boldsymbol{s}), \mathcal{P}_{\pi'}(\boldsymbol{s})$,

$$W_p(d_\pi)(\mathcal{P}_\pi(\boldsymbol{s}), \mathcal{P}_{\pi'}(\boldsymbol{s})) \leq \frac{c_R}{1 - c_T} D_{\mathrm{TV}}(\pi(\boldsymbol{s}), \pi'(\boldsymbol{s}))^{\frac{1}{p}}, \quad \forall \boldsymbol{s} \in \mathcal{S}. \tag{34}$$

*Proof.* First, recall from (6.11) of (Villani, 2008) that the 1-Wasserstein distance under the indicator function $\mathbf{1}[\boldsymbol{x} \neq \boldsymbol{x}']$ is equal to the total variation distance. Then,

$$
\begin{aligned}
W_1(\mathbf{1}[\boldsymbol{x} \neq \boldsymbol{x}'])(\mu_1, \mu_2) &= \min_{\omega \in \Omega} \|\mathbf{1}[\boldsymbol{x} \neq \boldsymbol{x}']\|_{1,\omega} = D_{\mathrm{TV}}(\mu_1, \mu_2) \\
&= \min_{\omega \in \Omega} \|\mathbf{1}[\boldsymbol{x} \neq \boldsymbol{x}']^p\|_{1,\omega}, \ \forall p \in [1, \infty) && (\text{since } \mathbf{1}[\boldsymbol{x} \neq \boldsymbol{x}'] = \mathbf{1}[\boldsymbol{x} \neq \boldsymbol{x}']^p) \\
&= \min_{\omega \in \Omega} \|\mathbf{1}[\boldsymbol{x} \neq \boldsymbol{x}']\|_{p,\omega}^p, \ \forall p \in [1, \infty) \\
&= W_p^p(\mathbf{1}[\boldsymbol{x} \neq \boldsymbol{x}'])(\mu_1, \mu_2), \ \forall p \in [1, \infty),
\end{aligned}
$$

which implies for all $p \in [1, \infty)$,

$$W_p(\mathbf{1}[\boldsymbol{x} \neq \boldsymbol{x}'])(\mu_1, \mu_2) = D_{\mathrm{TV}}(\mu_1, \mu_2)^{\frac{1}{p}}. \tag{35}$$

Next, note by Lemma 1 of Kemertas & Aumentado-Armstrong (2021), all bisimulation metrics are bounded above by $c_R/(1 - c_T)$ for $R \in [0, 1]$.

$$
\begin{aligned}
&W_p(d_\pi)(\mathcal{P}_\pi(\boldsymbol{s}), \mathcal{P}_{\pi'}(\boldsymbol{s})) \\
&= \frac{c_R}{1 - c_T} W_p\Big(\frac{d_\pi(1 - c_T)}{c_R}\Big)(\mathcal{P}_\pi(\boldsymbol{s}), \mathcal{P}_{\pi'}(\boldsymbol{s})) \\
&\leq \frac{c_R}{1 - c_T} W_p(\mathbf{1}[\boldsymbol{s} \neq \boldsymbol{s}'])(\mathcal{P}_\pi(\boldsymbol{s}), \mathcal{P}_{\pi'}(\boldsymbol{s})) \\
&= \frac{c_R}{1 - c_T} D_{\mathrm{TV}}(\mathcal{P}_\pi(\boldsymbol{s}), \mathcal{P}_{\pi'}(\boldsymbol{s}))^{\frac{1}{p}} && (\text{due to (35)}) \\
&\leq \frac{c_R}{1 - c_T} D_{\mathrm{TV}}(\pi(\boldsymbol{s}), \pi'(\boldsymbol{s}))^{\frac{1}{p}} && (\text{by Lemma A.5}) \quad \blacksquare
\end{aligned}
$$

**Lemma 3.6** (Comparing $\pi$-bisimulation metrics of different policies)**.** *Let* $\pi, \pi'$ *be a pair of policies and* $d_\pi^\sim, d_{\pi'}^\sim$ *corresponding* $\pi$*-bisimulation metrics given by* $p \in [1, \infty)$ *and* $\lambda = 0$*. The difference between* $d_\pi^\sim$ *and* $d_{\pi'}^\sim$ *is bounded by* $D_{\mathrm{TV}}^\infty(\pi, \pi') = \sup_{\boldsymbol{s} \in \mathcal{S}} D_{\mathrm{TV}}(\pi(\boldsymbol{s}), \pi'(\boldsymbol{s}))$*, the worst-case total variation distance of* $\pi$ *and* $\pi'$:

$$\|d_\pi^\sim - d_{\pi'}^\sim\|_\infty \leq \frac{2c_R}{(1 - c_T)^2} D_{\mathrm{TV}}^\infty(\pi, \pi')^{\frac{1}{p}}. \tag{18}$$

*Proof.*

$$
\begin{aligned}
&\Big|d_\pi(\boldsymbol{s}_i, \boldsymbol{s}_j) - d_{\pi'}(\boldsymbol{s}_i, \boldsymbol{s}_j)\Big| \\
&= \Big|c_R\Big(|R_\pi(\boldsymbol{s}_i) - R_\pi(\boldsymbol{s}_j)| - |R_{\pi'}(\boldsymbol{s}_i) - R_{\pi'}(\boldsymbol{s}_j)|\Big) \\
&\quad + c_T\Big(W_p(d_\pi)(\mathcal{P}_\pi(\boldsymbol{s}_i), \mathcal{P}_\pi(\boldsymbol{s}_j)) - W_p(d_{\pi'})(\mathcal{P}_{\pi'}(\boldsymbol{s}_i), \mathcal{P}_{\pi'}(\boldsymbol{s}_j))\Big)\Big| \\
&\leq c_R \underbrace{\Big||R_\pi(\boldsymbol{s}_i) - R_\pi(\boldsymbol{s}_j)| - |R_{\pi'}(\boldsymbol{s}_i) - R_{\pi'}(\boldsymbol{s}_j)|\Big|}_{\textcircled{1}} \\
&\quad + c_T \underbrace{\Big|W_p(d_\pi)(\mathcal{P}_\pi(\boldsymbol{s}_i), \mathcal{P}_\pi(\boldsymbol{s}_j)) - W_p(d_{\pi'})(\mathcal{P}_{\pi'}(\boldsymbol{s}_i), \mathcal{P}_{\pi'}(\boldsymbol{s}_j))\Big|}_{\textcircled{2}}
\end{aligned}
$$

$$
\begin{aligned}
\text{\textcircled{1}} &\le |R_\pi(\boldsymbol{s}_i) - R_{\pi'}(\boldsymbol{s}_i)| + |R_\pi(\boldsymbol{s}_j) - R_{\pi'}(\boldsymbol{s}_j)| \\
&\le D_{\mathrm{TV}}\Big(\pi(\boldsymbol{s}_i), \pi'(\boldsymbol{s}_i)\Big) + D_{\mathrm{TV}}\Big(\pi(\boldsymbol{s}_j), \pi'(\boldsymbol{s}_j)\Big) \qquad\qquad \text{(by Lemma A.6)} \\
&\le 2D_{\mathrm{TV}}^\infty(\pi, \pi').
\end{aligned}
$$

$$
\begin{aligned}
\text{\textcircled{2}} &\le W_p(d_\pi)(\mathcal{P}_\pi(\boldsymbol{s}_i), \mathcal{P}_{\pi'}(\boldsymbol{s}_i)) + W_p(d_\pi)(\mathcal{P}_\pi(\boldsymbol{s}_j)), \mathcal{P}_{\pi'}(\boldsymbol{s}_j)) + \|d_\pi - d_{\pi'}\|_\infty \qquad \text{(due to Corollary A.2)} \\
&\le \frac{c_R}{1 - c_T}\left( D_{\mathrm{TV}}\Big(\pi(\boldsymbol{s}_i), \pi'(\boldsymbol{s}_i)\Big)^{\frac{1}{p}} + D_{\mathrm{TV}}\Big(\pi(\boldsymbol{s}_j), \pi'(\boldsymbol{s}_j)\Big)^{\frac{1}{p}} \right) + c_T\|d_\pi - d_{\pi'}\|_\infty \quad \text{(due to Lemma A.7)} \\
&\le \frac{2c_R}{1 - c_T} D_{\mathrm{TV}}^\infty(\pi, \pi')^{\frac{1}{p}} + c_T\|d_\pi - d_{\pi'}\|_\infty
\end{aligned}
$$

Combining $\text{\textcircled{1}}$ and $\text{\textcircled{2}}$, and rearranging:

$$
\begin{aligned}
\Big|&d_\pi(\boldsymbol{s}_i, \boldsymbol{s}_j) - d_{\pi'}(\boldsymbol{s}_i, \boldsymbol{s}_j)\Big| \\
&\le 2c_R D_{\mathrm{TV}}^\infty(\pi, \pi') + \frac{2c_T c_R}{1 - c_T} D_{\mathrm{TV}}^\infty(\pi, \pi')^{\frac{1}{p}} + c_T\|d_\pi - d_{\pi'}\|_\infty \\
&\le 2c_R D_{\mathrm{TV}}^\infty(\pi, \pi')^{\frac{1}{p}} + \frac{2c_T c_R}{1 - c_T} D_{\mathrm{TV}}^\infty(\pi, \pi')^{\frac{1}{p}} + c_T\|d_\pi - d_{\pi'}\|_\infty \qquad \dagger \text{ (since } D_{\mathrm{TV}} \in [0,1]) \\
&\le \frac{2c_R}{1 - c_T} D_{\mathrm{TV}}^\infty(\pi, \pi')^{\frac{1}{p}} + c_T\|d_\pi - d_{\pi'}\|_\infty
\end{aligned}
$$

Taking a supremum over $\mathcal{S} \times \mathcal{S}$ on the LHS and rearranging,

$$
\|d_\pi - d_{\pi'}\|_\infty \le \frac{2c_R}{(1 - c_T)^2} D_{\mathrm{TV}}^\infty(\pi, \pi')^{\frac{1}{p}}. \qquad\qquad \blacksquare
$$

Note that step $\dagger$ serves only to simplify the final expression, but yields a loose bound for $p > 1$. Omitting $\dagger$ one obtains a better bound:

$$
\|d_\pi - d_{\pi'}\|_\infty \le \frac{2c_R}{(1 - c_T)}\Big( D_{\mathrm{TV}}^\infty(\pi, \pi') + \frac{c_T}{1 - c_T} D_{\mathrm{TV}}^\infty(\pi, \pi')^{\frac{1}{p}} \Big).
$$

As $c_T \to 1$ the second term dominates so that the loose bound tightens. Since $c_T$ is typically set to $\gamma$ and is close to 1 in practice, we use the simpler bound for convenience.

From Thm. 3.3 and the proof of Lemma 8 of Kemertas & Aumentado-Armstrong (2021), we have the following bound on approximation error for $V^\pi$ given an approximation of the $\pi$-bisimulation metric.

**Lemma A.8** (Approximating $V^\pi$ under metric learning error (Kemertas & Aumentado-Armstrong, 2021)). *Let $\widehat{d}_\pi$ be an approximation of $d_\pi^\sim$ such that $\|d_\pi^\sim - \widehat{d}_\pi\|_\infty \le \mathcal{E}$. If state abstraction function $\Phi$ is computed such that $\Phi(\boldsymbol{s}_i) = \Phi(\boldsymbol{s}_j) \Rightarrow \widehat{d}_\pi(\boldsymbol{s}_i, \boldsymbol{s}_j) \le 2\epsilon$, the following holds:*

$$
\|V^\pi - \widetilde{V}_\Phi^\pi\|_\infty \le \frac{2\epsilon + \mathcal{E}}{c_R(1 - \gamma)}. \tag{36}
$$

The following extends Thm. 4 of Kemertas & Aumentado-Armstrong (2021) to $(p, \zeta)$-Sinkhorn distances.

**Lemma A.9** (Decomposition of error sources in VFA for $(p, \zeta)$-Sinkhorn bisimulation metrics). *Consider a bounded approximate reward function $\widehat{R}$ and dynamics model $\widehat{\mathcal{P}}$ supported on a closed subset of $\mathcal{S}$. For any $p \ge 1$ and $\zeta \ge 0$, there exists a unique metric $d_{\pi,\widehat{\mathcal{P}}}$ such that*

$$
d_{\pi,\widehat{\mathcal{P}}}(\boldsymbol{s}_i, \boldsymbol{s}_j) = c_R|\widehat{R}_\pi(\boldsymbol{s}_i) - \widehat{R}_\pi(\boldsymbol{s}_i)| + c_T W_p^\zeta(d_{\pi,\widehat{\mathcal{P}}})(\widehat{\mathcal{P}}_\pi(\boldsymbol{s}_i), \widehat{\mathcal{P}}_\pi(\boldsymbol{s}_j)).
$$

*Furthermore, aggregation via a learned approximation $\widehat{d}_\pi$ of $d_{\pi,\widehat{\mathcal{P}}}$ yields the following bound for any $p \geq 1$:*

$$\|V^\pi - \widetilde{V}_\Phi^\pi\|_\infty \leq \frac{1}{c_R(1-\gamma)}\left(2\,\epsilon + \mathcal{E}_m + \frac{2c_R}{1-c_T}\mathcal{E}_R + \frac{2c_T}{1-c_T}\mathcal{E}_\mathcal{P}\right). \tag{37}$$

*where $\mathcal{E}_m := \|\widehat{d}_\pi - d_{\pi,\widehat{\mathcal{P}}}\|_\infty$ is the metric learning error, $\mathcal{E}_R := \|\widehat{R}_\pi - R_\pi\|_\infty$ is the reward approximation error, and $\mathcal{E}_\mathcal{P} := \sup_{s\in\mathcal{S}} W_p^\zeta(d_\pi)(\mathcal{P}_\pi(s), \widehat{\mathcal{P}}_\pi(s))$ is the state transition model error.*

*Proof.* Here, we use the following shorthand notation for the $\pi$-bisimulation distance between a pair of states:

$$d_\pi(s_i, s_j) = c_R|R_\pi^i - R_\pi^j| + c_T W_p^\zeta(d_\pi)(\mathcal{P}_\pi^i, \mathcal{P}_\pi^j).$$

First, note that the existence of $d_{\pi,\widehat{\mathcal{P}}}$ follows from Thm. 3 of Kemertas & Aumentado-Armstrong (2021) and Lemma 3.2. Next, we apply the triangle inequality to the error on the true metric $d_\pi$:

$$\begin{aligned}
&\|d_\pi - \widehat{d}_\pi\|_\infty \\
&\leq \|d_\pi - d_{\pi,\widehat{\mathcal{P}}}\|_\infty + \|\widehat{d}_\pi - d_{\pi,\widehat{\mathcal{P}}}\|_\infty &\text{(triangle inequality)} \\
&= \underbrace{\|d_\pi - d_{\pi,\widehat{\mathcal{P}}}\|_\infty}_{\text{①}} + \mathcal{E}_m &\text{(by definition of } \mathcal{E}_m)
\end{aligned}$$

Expanding ①:

$$\begin{aligned}
&\|d_\pi - d_{\pi,\widehat{\mathcal{P}}}\|_\infty \\
&= \sup_{i,j}\left\{\left|c_R\big(|R_\pi^i - R_\pi^j| - |\widehat{R}_\pi^i - \widehat{R}_\pi^j|\big) + c_T\Big(W_p^\zeta(d_\pi)(\mathcal{P}_\pi^i, \mathcal{P}_\pi^j) - W_p^\zeta(d_{\pi,\widehat{\mathcal{P}}})(\widehat{\mathcal{P}}_\pi^i, \widehat{\mathcal{P}}_\pi^j)\Big)\right|\right\} \\
&\leq \sup_{i,j}\left\{\left|c_R\big(|R_\pi^i - \widehat{R}_\pi^i| + |R_\pi^j - \widehat{R}_\pi^j|\big) + c_T\Big(W_p^\zeta(d_\pi)(\mathcal{P}_\pi^i, \mathcal{P}_\pi^j) - W_p^\zeta(d_{\pi,\widehat{\mathcal{P}}})(\widehat{\mathcal{P}}_\pi^i, \widehat{\mathcal{P}}_\pi^j)\Big)\right|\right\} \\
&\leq 2c_R\mathcal{E}_R + c_T\sup_{i,j}\left|\Big(W_p^\zeta(d_\pi)(\mathcal{P}_\pi^i, \mathcal{P}_\pi^j) - W_p^\zeta(d_{\pi,\widehat{\mathcal{P}}})(\widehat{\mathcal{P}}_\pi^i, \widehat{\mathcal{P}}_\pi^j)\Big)\right| &\text{(by definition of } \mathcal{E}_R) \\
&\leq 2c_R\mathcal{E}_R + c_T\sup_{i,j}\Big(W_p^\zeta(d_\pi)(\mathcal{P}_\pi^i, \widehat{\mathcal{P}}_\pi^i) + W_p^\zeta(d_\pi)(\mathcal{P}_\pi^j, \widehat{\mathcal{P}}_\pi^j) + \|d_\pi - d_{\pi,\widehat{\mathcal{P}}}\|_\infty\Big) &\text{(by Corollary A.2)} \\
&\leq 2c_R\mathcal{E}_R + 2c_T\mathcal{E}_\mathcal{P} + c_T\|d_\pi - d_{\pi,\widehat{\mathcal{P}}}\|_\infty. &\text{(by definition of } \mathcal{E}_\mathcal{P})
\end{aligned}$$

Rearranging,

$$\|d_\pi - d_{\pi,\widehat{\mathcal{P}}}\|_\infty \leq \frac{2c_R}{1-c_T}\mathcal{E}_R + \frac{2c_T}{1-c_T}\mathcal{E}_\mathcal{P}.$$

Plugging ① back in,

$$\|d_\pi - \widehat{d}_\pi\|_\infty \leq \mathcal{E}_m + \frac{2c_R}{1-c_T}\mathcal{E}_R + \frac{2c_T}{1-c_T}\mathcal{E}_\mathcal{P}.$$

The result follows from Lemma A.8. ∎

Now, we rephrase API bounds from Bertsekas (2018a) before proving Thm. 3.5.

**Lemma A.10** (Propositions 2.4.3 and 2.4.5 of (Bertsekas, 2018a)). *Consider an API algorithm that generates policies $\{\pi_k\}_{k\in\mathbb{N}}$ and functions $\{V_k\}_{k\in\mathbb{N}}$ in $\mathcal{B}(\mathcal{S})$ with policy evaluation error $\|V_k - V^{\pi_k}\|_\infty \leq \delta_{\mathrm{PE}}$ and approximate greedy updates with error $\|T_{\pi_{k+1}}V_k - TV_k\|_\infty \leq \delta_{\mathrm{GI}}$. If the sequence converges to a policy $\overline{\pi}$,*

$$\|V^{\overline{\pi}} - V^*\|_\infty \leq \frac{\delta_{\mathrm{GI}} + 2\gamma\delta_{\mathrm{PE}}}{1-\gamma}.$$

*Otherwise, the sequence $\{\pi_k\}_{k\in\mathbb{N}}$ has the limiting bound,*

$$\limsup_{k\to\infty}\|V^{\pi_k} - V^*\|_\infty \leq \frac{\delta_{\mathrm{GI}} + 2\gamma\delta_{\mathrm{PE}}}{(1-\gamma)^2}.$$

**Theorem 3.5** (API with $\pi$-bisimulation). *Let $c_R = 1$, $c_T = \gamma$ and $\{\pi_k\}_{k\in\mathbb{N}}$ be a sequence of policies generated with the following updates per step, where $d_0 = \mathbf{0} \in \mathfrak{met}(\mathcal{S})$, and $\epsilon \geq 0$ and $n \in \mathbb{N}_+$ are algorithm parameters. Let $c_n = \gamma^n/(1-\gamma)$, and consider for any $p \geq 1$ and $\zeta \geq 0$:*

$$\widehat{d}_{\pi_k} \leftarrow \mathcal{F}_{\pi_k}^{(n)}(d_0) \tag{12}$$

$$\widetilde{\mathcal{S}}, \Phi_k \leftarrow \text{HardAggregation}(\mathcal{S}, \widehat{d}_{\pi_k}, \epsilon) \tag{13}$$

$$\widetilde{V}^{\pi_k} \leftarrow \text{PolicyEvaluation}(\widetilde{\mathcal{S}}, \pi_k) \tag{14}$$

$$\pi_{k+1} \leftarrow \text{GreedyImprovement}(\widetilde{V}_{\Phi_k}^{\pi_k}, \delta), \tag{15}$$

*and $\widetilde{V}_{\Phi_k}^{\pi_k} \in \mathcal{B}(\mathcal{S})$ is the composition of $\widetilde{V}^{\pi_k}$ and $\Phi_k$. If the sequence $\{\pi_k\}_{k\in\mathbb{N}}$ converges to a policy $\overline{\pi}$, we have*

$$\|V^{\overline{\pi}} - V^*\|_\infty \leq \frac{\delta}{1-\gamma} + \frac{2\gamma(2\epsilon + c_n)}{(1-\gamma)^2}. \tag{16}$$

*Otherwise, it has the following limiting bound,*

$$\limsup_{k\to\infty} \|V^{\pi_k} - V^*\|_\infty \leq \frac{\delta}{(1-\gamma)^2} + \frac{2\gamma(2\epsilon + c_n)}{(1-\gamma)^3}. \tag{17}$$

*Proof.* First, we observe that,

$$\begin{aligned}
&\|\widehat{d}_{\pi_k} - d_{\pi_k}\|_\infty \\
&= \|\mathcal{F}_{\pi_k}^{(n)}(d_0) - d_{\pi_k}\|_\infty \\
&\leq \frac{c_T^n}{1-c_T}\|\mathcal{F}_{\pi_k}(d_0) - d_0\|_\infty &&\text{(by the Banach fixed-point theorem and Thm. 3.3)} \\
&= c_n\|\mathcal{F}_{\pi_k}(d_0)\|_\infty &&\text{(since } d_0 = \mathbf{0}) \\
&= c_n \sup_{i,j} |R_\pi(\boldsymbol{s}_i) - R_\pi(\boldsymbol{s}_j)| &&\text{(since } d_0 = \mathbf{0} \Rightarrow W_p(d_0) = 0) \\
&\leq c_n, \forall k \in \mathbb{N}. &&\text{(since } R \in [0,1])
\end{aligned}$$

Then, by Lemma A.8,

$$\|V^{\pi_k} - \widetilde{V}_{\Phi_k}^{\pi_k}\|_\infty \leq \frac{2\epsilon + c_n}{1-\gamma}, \forall k \in \mathbb{N}.$$

The result follows from Lemma A.10 with $\delta_{\text{GI}} = \delta$ and $\delta_{\text{PE}} = \frac{2\epsilon + c_n}{1-\gamma}$. ∎

To prove Thm. 3.8, we write an analogue of Lemma A.10 that does not assume a fixed bound on PE error, but a sequence of policy evaluation errors $\delta_{\text{PE},k}$ and greedy improvement errors $\delta_{\text{GI},k}$ that have finite limiting bounds. The following is a slight variation of Lemma A.10, which is stronger as it considers asymptotic bounds on said errors rather than the maximum error over all $k$.

**Lemma A.11** (A stronger API bound (Bertsekas, 2018a)). *Consider an API algorithm that generates policies $\{\pi_k\}_{k\in\mathbb{N}}$ and functions $\{V_k\}_{k\in\mathbb{N}}$ in $\mathcal{B}(\mathcal{S})$ with policy evaluation error $\|V_k - V^{\pi_k}\|_\infty \leq \delta_{\text{PE},k}$ and approximate greedy updates with error $\|T_{\pi_{k+1}}V_k - TV_k\|_\infty \leq \delta_{\text{GI},k}$. The sequence $\{\pi_k\}_{k\in\mathbb{N}}$ has the limiting bound,*

$$\limsup_{k\to\infty} \|V^{\pi_k} - V^*\|_\infty \leq \frac{\limsup_{k\to\infty} \delta_{\text{GI},k} + 2\gamma\delta_{\text{PE},k}}{(1-\gamma)^2}.$$

*Proof.* From Prop. 2.4.4 of (Bertsekas, 2018a), given $\|V_k - V^{\pi_k}\|_\infty \leq \delta_{\text{PE},k}$ and $\|T_{\pi_{k+1}}V_k - TV_k\|_\infty \leq \delta_{\text{GI},k}$,

$$\|V^{\pi_{k+1}} - V^*\|_\infty \leq \gamma\|V^{\pi_k} - V^*\|_\infty + \frac{\delta_{\text{GI},k} + 2\gamma\delta_{\text{PE},k}}{1-\gamma}. \tag{38}$$

The result follows by simply taking a $\limsup_{k\to\infty}$ on both sides. ∎

**Lemma 3.7** (Generalized API($\alpha$) bounds). *Let $V \in \mathcal{B}(\mathcal{S})$ and policies $\pi, \pi', \pi_g$ satisfy the following:*

$$\|V^\pi - V\|_\infty \le \delta_{\mathrm{PE}}$$
$$\|T_{\pi_g} V - TV\|_\infty \le \delta_{\mathrm{GI}}$$
$$\pi' = \alpha \pi_g + (1 - \alpha)\pi,$$

*where $\alpha \in [0, 1]$. Then,*

$$\|V^{\pi'} - V^*\|_\infty \le (1 - \alpha + \alpha\gamma)\|V^\pi - V^*\|_\infty + \alpha \frac{\delta_{\mathrm{GI}} + 2\gamma\delta_{\mathrm{PE}}}{1 - \gamma}. \tag{19}$$

*Next, consider an API($\alpha$) algorithm that generates a sequence of policies $\{\pi_k\}_{k \in \mathbb{N}}$ via functions $\{V_k\}_{k \in \mathbb{N}}$ with policy evaluation error $\|V^{\pi_k} - V_k\|_\infty \le \delta_{\mathrm{PE},k}$, approximate greedy updates with error $\|T_{\pi_{g,k}} V_k - TV_k\|_\infty \le \delta_{\mathrm{GI,k}}$ and policy updates $\pi_{k+1} \leftarrow \alpha\pi_{g,k} + (1 - \alpha)\pi_k$. For any $\alpha \in (0, 1]$,*

$$\limsup_{k \to \infty} \|V^{\pi_k} - V^*\|_\infty \le \frac{\limsup_{k \to \infty} \delta_{\mathrm{GI},k} + 2\gamma\delta_{\mathrm{PE},k}}{(1 - \gamma)^2}. \tag{20}$$

*Proof.* Let $e = \delta_{\mathrm{GI}} + 2\gamma\delta_{\mathrm{PE}}$. First, we note that $T_{\pi'} V = \alpha T_{\pi_g} V + (1 - \alpha) T_\pi V$ for all $V \in \mathcal{B}(\mathcal{S})$, and prove the following:

$$\sup_{s \in \mathcal{S}} \{V^\pi(s) - V^{\pi'}(s)\} \le \frac{\alpha e}{1 - \gamma}. \tag{39}$$

$$V^\pi - V^{\pi'}$$
$$= T_\pi V^\pi - T_{\pi'} V^{\pi'}$$
$$= T_\pi V^\pi - \alpha T_{\pi_g} V^{\pi'} - (1 - \alpha) T_\pi V^{\pi'}$$
$$= \alpha(T_\pi V^\pi - T_{\pi_g} V^{\pi'}) + (1 - \alpha)(T_\pi V^\pi - T_\pi V^{\pi'})$$
$$\le \alpha(T_\pi V^\pi - T_{\pi_g} V^{\pi'}) + (1 - \alpha)\gamma \sup_{s \in \mathcal{S}} \{V^\pi(s) - V^{\pi'}(s)\}$$
$$= \alpha(T_\pi V^\pi - T_\pi V + T_\pi V - T_{\pi_g} V^{\pi'}) + (1 - \alpha)\gamma \sup_{s \in \mathcal{S}} \{V^\pi(s) - V^{\pi'}(s)\}$$
$$\le \alpha(\gamma\delta_{\mathrm{PE}} + T_\pi V - T_{\pi_g} V^{\pi'}) + (1 - \alpha)\gamma \sup_{s \in \mathcal{S}} \{V^\pi(s) - V^{\pi'}(s)\}$$
$$\le \alpha(\gamma\delta_{\mathrm{PE}} + TV - T_{\pi_g} V^{\pi'}) + (1 - \alpha)\gamma \sup_{s \in \mathcal{S}} \{V^\pi(s) - V^{\pi'}(s)\} \qquad \text{(since } TV \ge T_\pi V \text{ for all } \pi.)$$
$$= \alpha(\gamma\delta_{\mathrm{PE}} + TV - T_{\pi_g} V + T_{\pi_g} V - T_{\pi_g} V^{\pi'}) + (1 - \alpha)\gamma \sup_{s \in \mathcal{S}} \{V^\pi(s) - V^{\pi'}(s)\}$$
$$\le \alpha(\gamma\delta_{\mathrm{PE}} + \delta_{\mathrm{GI}} + T_{\pi_g} V - T_{\pi_g} V^{\pi'}) + (1 - \alpha)\gamma \sup_{s \in \mathcal{S}} \{V^\pi(s) - V^{\pi'}(s)\}$$
$$= \alpha(\gamma\delta_{\mathrm{PE}} + \delta_{\mathrm{GI}} + T_{\pi_g} V - T_{\pi_g} V^\pi + T_{\pi_g} V^\pi - T_{\pi_g} V^{\pi'}) + (1 - \alpha)\gamma \sup_{s \in \mathcal{S}} \{V^\pi(s) - V^{\pi'}(s)\}$$
$$\le \alpha(2\gamma\delta_{\mathrm{PE}} + \delta_{\mathrm{GI}} + T_{\pi_g} V^\pi - T_{\pi_g} V^{\pi'}) + (1 - \alpha)\gamma \sup_{s \in \mathcal{S}} \{V^\pi(s) - V^{\pi'}(s)\}$$
$$= \alpha(e + T_{\pi_g} V^\pi - T_{\pi_g} V^{\pi'}) + (1 - \alpha)\gamma \sup_{s \in \mathcal{S}} \{V^\pi(s) - V^{\pi'}(s)\}$$
$$\le \alpha(e + \gamma \sup_{s \in \mathcal{S}} \{V^\pi(s) - V^{\pi'}(s)\}) + (1 - \alpha)\gamma \sup_{s \in \mathcal{S}} \{V^\pi(s) - V^{\pi'}(s)\}$$
$$= \alpha e + \gamma \sup_{s \in \mathcal{S}} \{V^\pi(s) - V^{\pi'}(s)\}.$$

By taking a supremum on the LHS and rearranging, we obtain (39). Now, we prove (19).

$$\begin{aligned}
V^* - V^{\pi'} &= TV^* - T_{\pi'}V^{\pi'} \\
&= TV^* - \alpha T_{\pi_g}V^{\pi'} - (1-\alpha)T_\pi V^{\pi'} \\
&= \alpha \underbrace{(TV^* - T_{\pi_g}V^{\pi'})}_{\textcircled{1}} + (1-\alpha)\underbrace{(TV^* - T_\pi V^{\pi'})}_{\textcircled{2}}.
\end{aligned}$$

$$\begin{aligned}
\textcircled{1} &= TV^* - T_{\pi_g}V^\pi + T_{\pi_g}V^\pi - T_{\pi_g}V^{\pi'} \\
&\le TV^* - T_{\pi_g}V^\pi + \gamma \sup_{s\in\mathcal{S}}\{V^\pi(s) - V^{\pi'}(s)\} \\
&\le TV^* - T_{\pi_g}V^\pi + \frac{\gamma\alpha e}{1-\gamma} & \text{(due to (39))}\\
&= TV^* - T_{\pi_g}V + T_{\pi_g}V - T_{\pi_g}V^\pi + \frac{\gamma\alpha e}{1-\gamma} \\
&\le TV^* - T_{\pi_g}V + \gamma\delta_{\text{PE}} + \frac{\gamma\alpha e}{1-\gamma} \\
&= TV^* - TV^\pi + TV^\pi - T_{\pi_g}V + \gamma\delta_{\text{PE}} + \frac{\gamma\alpha e}{1-\gamma} \\
&\le \gamma\|V^* - V^\pi\|_\infty + TV^\pi - T_{\pi_g}V + \gamma\delta_{\text{PE}} + \frac{\gamma\alpha e}{1-\gamma} \\
&= \gamma\|V^* - V^\pi\|_\infty + TV^\pi - TV + TV - T_{\pi_g}V + \gamma\delta_{\text{PE}} + \frac{\gamma\alpha e}{1-\gamma} \\
&\le \gamma\|V^* - V^\pi\|_\infty + 2\gamma\delta_{\text{PE}} + \delta_{\text{GI}} + \frac{\gamma\alpha e}{1-\gamma} \\
&= \gamma\|V^* - V^\pi\|_\infty + e + \frac{\gamma\alpha e}{1-\gamma}.
\end{aligned}$$

$$\begin{aligned}
\textcircled{2} &= V^* - V^\pi + V^\pi - T_\pi V^{\pi'} \\
&\le \|V^* - V^\pi\|_\infty + V^\pi - T_\pi V^{\pi'} \\
&= \|V^* - V^\pi\|_\infty + T_\pi V^\pi - T_\pi V^{\pi'} \\
&\le \|V^* - V^\pi\|_\infty + \gamma \sup_{s\in\mathcal{S}}\{V^\pi(s) - V^{\pi'}(s)\} \\
&\le \|V^* - V^\pi\|_\infty + \frac{\gamma\alpha e}{1-\gamma}. & \text{(due to (39))}
\end{aligned}$$

Combining the upper bounds of $\textcircled{1}$ and $\textcircled{2}$,

$$\begin{aligned}
V^* - V^{\pi'} &\le \alpha\Big(\gamma\|V^* - V^\pi\|_\infty + e + \frac{\gamma\alpha e}{1-\gamma}\Big) + (1-\alpha)\Big(\|V^* - V^\pi\|_\infty + \frac{\gamma\alpha e}{1-\gamma}\Big) \\
&= (1-\alpha+\alpha\gamma)\|V^* - V^\pi\|_\infty + \alpha\left(e + \frac{\gamma e}{1-\gamma}\right) \\
&= (1-\alpha+\alpha\gamma)\|V^* - V^\pi\|_\infty + \frac{\alpha e}{1-\gamma},
\end{aligned}$$

which is equivalent to (19). Setting $\pi' = \pi_{k+1}$ and $\pi = \pi_k$, and taking a lim sup on both sides of (19):

$$\begin{aligned}
&\limsup_{k\to\infty}\|V^{\pi_{k+1}} - V^*\|_\infty \le \limsup_{k\to\infty}(1-\alpha+\alpha\gamma)\|V^{\pi_k} - V^*\|_\infty + \alpha\frac{\limsup_{k\to\infty}\delta_{\text{GI,k}} + 2\gamma\delta_{\text{PE,k}}}{1-\gamma} \\
\Rightarrow\quad & \alpha(1-\gamma)\limsup_{k\to\infty}\|V^{\pi_k} - V^*\|_\infty \le \alpha\frac{\limsup_{k\to\infty}\delta_{\text{GI,k}} + 2\gamma\delta_{\text{PE,k}}}{1-\gamma} \\
\Rightarrow\quad & \limsup_{k\to\infty}\|V^{\pi_k} - V^*\|_\infty \le \frac{\limsup_{k\to\infty}\delta_{\text{GI,k}} + 2\gamma\delta_{\text{PE,k}}}{(1-\gamma)^2}. \qquad\blacksquare
\end{aligned}$$

**Theorem 3.8** (API($\alpha$) with $\pi$-bisimulation). *Under the same conventions as Thm. 3.5, let $n > \log(\frac{1-\gamma}{1+\gamma})/\log(\gamma)$ and $\bar{c}_n = (1+\gamma)c_n$. Given $\alpha = \left(\overline{\alpha}(1-\bar{c}_n)(1-\gamma)/2\right)^p$ for some $\overline{\alpha} \in (0,1]$, $\lambda = 0$ and any $p \in [1, \infty)$:*

$$\widehat{d}_{\pi_k} \leftarrow \mathcal{F}_{\pi_k}^{(n)}(\widehat{d}_{\pi_{k-1}}) \tag{21}$$

$$\widetilde{\mathcal{S}}, \Phi_k \leftarrow \text{HardAggregation}(\mathcal{S}, \widehat{d}_{\pi_k}, \epsilon) \tag{22}$$

$$\widetilde{V}^{\pi_k} \leftarrow \text{PolicyEvaluation}(\widetilde{\mathcal{S}}, \pi_k) \tag{23}$$

$$\pi_g \leftarrow \text{GreedyImprovement}(\widetilde{V}_{\Phi_k}^{\pi_k}, \delta) \tag{24}$$

$$\pi_{k+1} \leftarrow (1-\alpha)\pi_k + \alpha\pi_g. \tag{25}$$

*The sequence $\{\pi_k\}_{k\in\mathbb{N}}$ has the following limiting bound,*

$$\limsup_{k\to\infty}\|V^{\pi_k} - V^*\|_\infty \leq \frac{\delta}{(1-\gamma)^2} + \frac{2\gamma(2\epsilon + \overline{\alpha}c_n)}{(1-\gamma)^3}. \tag{26}$$

*Proof.* First, note that by Corollary A.4, we have $D_{\text{TV}}^\infty(\pi_{k+1}, \pi_k) \leq \alpha$. Now, we define the sequence of metric learning errors $\{\mathcal{E}_k\}_{k\in\mathbb{N}}$,

$$\begin{aligned}
\mathcal{E}_k &= \|\widehat{d}_{\pi_k} - d_{\pi_k}\|_\infty \\
&= \|\mathcal{F}_{\pi_k}^{(n)}(\widehat{d}_{\pi_{k-1}}) - d_{\pi_k}\|_\infty \\
&\leq c_n\|\mathcal{F}_{\pi_k}(\widehat{d}_{\pi_{k-1}}) - \widehat{d}_{\pi_{k-1}}\|_\infty && \text{(by the Banach fixed-point theorem)} \\
&\leq c_n\left(\|\mathcal{F}_{\pi_k}(\widehat{d}_{\pi_{k-1}}) - d_{\pi_{k-1}}\|_\infty + \|\widehat{d}_{\pi_{k-1}} - d_{\pi_{k-1}}\|_\infty\right) \\
&= c_n\Big(\underbrace{\|\mathcal{F}_{\pi_k}(\widehat{d}_{\pi_{k-1}}) - d_{\pi_{k-1}}\|_\infty}_{\text{①}} + \mathcal{E}_{k-1}\Big)
\end{aligned}$$

Using shorthand notation $d_\pi = c_R|R_\pi^i - R_\pi^j| + c_T W_p(d_\pi)(\mathcal{P}_\pi^i, \mathcal{P}_\pi^j)$ for the $\pi$-bisimulation distance between a pair of states $(\boldsymbol{s}_i, \boldsymbol{s}_j)$, and noting $c_R = 1$ by given,

$$\begin{aligned}
\text{①} &= \sup_{i,j}\left||R_{\pi_k}^i - R_{\pi_k}^j| - |R_{\pi_{k-1}}^i - R_{\pi_{k-1}}^j| + c_T\left(W_p(\widehat{d}_{\pi_{k-1}})(\mathcal{P}_{\pi_k}^i, \mathcal{P}_{\pi_k}^j) - W_p(d_{\pi_{k-1}})(\mathcal{P}_{\pi_{k-1}}^i, \mathcal{P}_{\pi_{k-1}}^j)\right)\right| \\
&\leq \sup_{i,j}\left||R_{\pi_k}^i - R_{\pi_{k-1}}^i| + |R_{\pi_k}^j - R_{\pi_{k-1}}^j| + c_T\left(W_p(\widehat{d}_{\pi_{k-1}})(\mathcal{P}_{\pi_k}^i, \mathcal{P}_{\pi_k}^j) - W_p(d_{\pi_{k-1}})(\mathcal{P}_{\pi_{k-1}}^i, \mathcal{P}_{\pi_{k-1}}^j)\right)\right| \\
&\leq 2D_{\text{TV}}^\infty(\pi_k, \pi_{k-1}) + c_T\sup_{i,j}\left|W_p(\widehat{d}_{\pi_{k-1}})(\mathcal{P}_{\pi_k}^i, \mathcal{P}_{\pi_k}^j) - W_p(d_{\pi_{k-1}})(\mathcal{P}_{\pi_{k-1}}^i, \mathcal{P}_{\pi_{k-1}}^j)\right| && \text{(by Lemma A.6)} \\
&\leq 2\alpha + c_T\sup_{i,j}\left|W_p(\widehat{d}_{\pi_{k-1}})(\mathcal{P}_{\pi_k}^i, \mathcal{P}_{\pi_k}^j) - W_p(d_{\pi_{k-1}})(\mathcal{P}_{\pi_{k-1}}^i, \mathcal{P}_{\pi_{k-1}}^j)\right| && \text{(since } D_{\text{TV}}^\infty(\pi_k, \pi_{k-1}) \leq \alpha) \\
&\leq 2\alpha + c_T\left(W_p(d_{\pi_{k-1}})(\mathcal{P}_{\pi_{k-1}}^i, \mathcal{P}_{\pi_k}^i) + W_p(d_{\pi_{k-1}})(\mathcal{P}_{\pi_{k-1}}^j, \mathcal{P}_{\pi_k}^j) + \|\widehat{d}_{\pi_{k-1}} - d_{\pi_{k-1}}\|_\infty\right) && \text{(by Cor. A.2)} \\
&\leq 2\alpha + \frac{2c_T}{1-c_T}D_{\text{TV}}^\infty(\pi_k, \pi_{k-1})^{\frac{1}{p}} + c_T\|\widehat{d}_{\pi_{k-1}} - d_{\pi_{k-1}}\|_\infty && \text{(by Lemma A.7)} \\
&\leq 2\alpha^{\frac{1}{p}} + \frac{2c_T\alpha^{\frac{1}{p}}}{1-c_T} + c_T\|\widehat{d}_{\pi_{k-1}} - d_{\pi_{k-1}}\|_\infty && \text{(since } \alpha \in [0,1] \text{ and } D_{\text{TV}}^\infty(\pi_k, \pi_{k-1}) \leq \alpha) \\
&\leq \frac{2\alpha^{\frac{1}{p}}}{1-c_T} + c_T\mathcal{E}_{k-1}
\end{aligned}$$

Plugging $\textcircled{1}$ back in,

$$\mathcal{E}_k \leq c_n \left( \frac{2\alpha^{\frac{1}{p}}}{1 - c_T} + (1 + c_T)\mathcal{E}_{k-1} \right)$$

$$= \frac{2\alpha^{\frac{1}{p}} c_n}{1 - c_T} + \bar{c}_n \mathcal{E}_{k-1}$$

If $\bar{c}_n = (1 + c_T)c_n < 1$, i.e., $n > \log(\frac{1-c_T}{1+c_T})/\log(c_T)$, we take a limit superior to obtain:

$$\limsup_{k\to\infty} \mathcal{E}_k \leq \frac{2\alpha^{\frac{1}{p}} c_n}{(1 - \bar{c}_n)(1 - c_T)}$$

$$= \bar{\alpha}c_n,$$

where we have defined $\bar{\alpha} = \frac{2\alpha^{\frac{1}{p}}}{(1-\bar{c}_n)(1-c_T)}$. By Lemma A.8, we have,

$$\limsup_{k\to\infty} \|V^{\pi_k} - \widetilde{V}_{\Phi_k}^{\pi_k}\|_\infty \leq \frac{2\epsilon + \bar{\alpha}c_n}{1 - \gamma}.$$

Then, (26) holds due to (20) of Lemma 3.7 with $\delta_{\text{GI,k}} \leq \delta$, $\forall k$ and $\delta_{\text{PE},k} = \|V^{\pi_k} - \widetilde{V}_{\Phi_k}^{\pi_k}\|_\infty$. $\blacksquare$

## B    Background: State Aggregation Methods

The idea of reducing a large-scale dynamic programming problem into a smaller one via abstractions (or partitions) has a rich history going back many decades (Fox, 1973; Whitt, 1978; Mendelssohn, 1982; Bertsekas & Castanon, 1989; Singh et al., 1995; Dean & Givan, 1997; Dean et al., 1997). Broadly, state aggregation approaches in RL can be grouped into two categories: pre-specified and adaptive. Often, pre-specified approaches either compute an aggregation based on transition probabilities and reward functions as in bisimulation (Givan et al., 2003) or assume a priori knowledge about the environment (e.g., the optimal value function). For instance, given some known function $f : \mathcal{S} \times \mathcal{A} \to \mathbb{R}$, instead of (1) we may write:

$$\Phi(\boldsymbol{s}_i) = \Phi(\boldsymbol{s}_j) \Rightarrow |f(\boldsymbol{s}_i, \boldsymbol{a}) - f(\boldsymbol{s}_j, \boldsymbol{a})| \leq \eta, \forall \boldsymbol{a} \in \mathcal{A}, \tag{40}$$

which results in $\eta$-abstractions of Abel et al. (2016). When $f = Q^*$ and $\eta = 0$, the resulting abstraction is called a "$Q^*$-irrelevance abstraction" under the unifying framework of Li et al. (2006). McCallum (1996) introduced the Utile Distinction Test (in the context of partially observable MDPs), which aggregates states that have the same optimal action and the same state-action value for said action. Hostetler et al. (2014) investigated a similar, more general abstraction in the context of Monte Carlo Tree Search to reduce the stochastic branching factor of large MDPs. Jong & Stone (2005) devised an abstraction discovery approach based on statistical hypothesis testing and policy relevance, illustrated its utility in knowledge transfer to different domains and discussed its connections to hierarchical RL (i.e., temporal abstraction). Duan et al. (2019) developed a soft aggregation algorithm based on the spectral decomposition of a simulation-based empirical transition matrix. Bertsekas (2018b) recently surveyed feature-based aggregation methods (for an early example, see Tsitsiklis & Van Roy (1996)) and discussed their use in API. Van Roy (2006) analyzed approximate value iteration (AVI) under state aggregation. In general, whenever VFA is done via piece-wise constant function approximators, state aggregation is implicit among states that are assigned the same value.

More closely related to this work, adaptive approaches simultaneously improve a policy and learn an efficient abstraction that changes as the algorithm runs. A notable early example is the work of Bertsekas & Castanon (1989), which adaptively aggregates states to minimize the variation in Bellman residuals per partition. Baras & Borkar (2000) developed a simulation-based actor-critic approach that alternates between frequent updates to a linear approximation of the value function and infrequent updates to abstractions based on a clustering in the range of estimated values. Ortner (2013) developed an online aggregation algorithm based on the UCRL algorithm of Auer et al. (2008) and provided a regret analysis. The approach of Sinclair et al. (2019) was

similarly motivated, but focused on $Q$-learning. Chen et al. (2021) recently proposed an aggregation-based AVI algorithm that combines infrequent Bellman updates over the ground space with frequent updates over a reduced space that is computed via value-based aggregation. Differently, our approach does not require any value iteration updates in the ground space, which may be continuous.

## C  Sharpness of the Sinkhorn Distance as a Wasserstein Distance Upper Bound

The Sinkhorn distance forms an upper bound on the Wasserstein distance due to an entropy-constraint imposed on the cost minimization problem. Guided by information-theoretic intuition, we perform empirical tests to investigate the quality of this upper bound as a function of the underlying marginal distributions $(\mu_1, \mu_2)$. First, we highlight the intuition with the following lemma.

**Lemma C.1** (A condition for equality of $W_p$ and $W_p^\zeta$). *Let $\mathcal{H}(\mu)$ denote the Shannon entropy of a random variable with law $\mu$. Under the same conventions as (7),*

$$\zeta^{-1} \geq \min(\mathcal{H}(\mu_1), \mathcal{H}(\mu_2)) \Rightarrow W_p^\zeta(d)(\mu_1, \mu_2) = W_p(d)(\mu_1, \mu_2). \tag{41}$$

*Proof.* Consider a joint distribution $\omega$ with marginals $\mu_1$ and $\mu_2$. Recall the following information-theoretic inequalities (Cover, 1999):

$$\mathcal{H}(\omega) \leq \mathcal{H}(\mu_1) + \mathcal{H}(\mu_2) \tag{42}$$
$$\mathcal{H}(\omega) \geq \mathcal{H}(\mu_1). \tag{43}$$

By symmetry of (43),

$$\max(\mathcal{H}(\mu_1), \mathcal{H}(\mu_2)) \leq \mathcal{H}(\omega) \leq \mathcal{H}(\mu_1) + \mathcal{H}(\mu_2) \tag{44}$$
$$\Rightarrow \max(\mathcal{H}(\mu_1), \mathcal{H}(\mu_2)) \leq \mathcal{H}(\omega) \leq \max(\mathcal{H}(\mu_1), \mathcal{H}(\mu_2)) + \min(\mathcal{H}(\mu_1), \mathcal{H}(\mu_2)). \tag{45}$$

That is, the range of allowed entropy values for a joint distribution $\omega$ with marginals $\mu_1$ and $\mu_2$ is determined by the minimum entropy of the two distributions. For example, suppose $\mu_1(x_i) = 1$ for some $x_i$ in a finite space and 0 elsewhere, so that $\mathcal{H}(\mu_1) = 0$. The feasible set of transport plans between $\mu_1$ and $\mu_2$ collapses to a single joint distribution that moves all the probability mass at $x_i$ to match the distribution of $\mu_2$. Indeed, in this case we have $\mathcal{H}(\omega) = \mathcal{H}(\mu_1) + \mathcal{H}(\mu_2) = \mathcal{H}(\mu_2)$ with equality for both the lower and upper bounds shown in (45).

Now, consider the relative entropy-constrained set of joints $\Omega(\zeta)$ from (7):

$$D_{\mathrm{KL}}(\omega \,||\, \mu_1 \otimes \mu_2) \leq \zeta^{-1}$$
$$\Rightarrow -\mathcal{H}(\omega) + \mathcal{H}(\mu_1) + \mathcal{H}(\mu_2) \leq \zeta^{-1}$$
$$\Rightarrow \mathcal{H}(\omega) \geq \mathcal{H}(\mu_1) + \mathcal{H}(\mu_2) - \zeta^{-1}$$
$$\Rightarrow \mathcal{H}(\omega) \geq \max(\mathcal{H}(\mu_1), \mathcal{H}(\mu_2)) + \min(\mathcal{H}(\mu_1), \mathcal{H}(\mu_2)) - \zeta^{-1}.$$

Given the information-theoretic lower bound in (45) that readily applies to all $\omega \in \Omega$, we conclude that whenever $\zeta^{-1} \geq \min(\mathcal{H}(\mu_1), \mathcal{H}(\mu_2))$ we have $\Omega = \Omega(\zeta)$, which implies equality between $W_p$ and $W_p^\zeta$. ∎

As discussed in the proof above, the range of allowed entropy values $\mathcal{H}(\omega)$ shrinks with smaller minimum entropy $\min(\mathcal{H}(\mu_1), \mathcal{H}(\mu_2))$. Consequently, as $\min(\mathcal{H}(\mu_1), \mathcal{H}(\mu_2)) \to 0$ we have $\Omega(\zeta) \to \Omega$ for all $\zeta$ due to (41). We suspect that the converse may be true; that with increasing $\min(\mathcal{H}(\mu_1), \mathcal{H}(\mu_2))$ the feasible set $\Omega(\zeta)$ might be a smaller subset of $\Omega$, which would imply that the quality of the Sinkhorn distance as a Wasserstein distance upper bound degrades. We perform empirical tests to compare the $W_1^\lambda$ distance to a stronger upper bound on the 1-Wasserstein distance computed via a smaller $\lambda' = 0.02 < \lambda$. We randomly sample probability vectors $\mu_1, \mu_2$ from the 31-simplex and ensure $\mathcal{H}(\mu_1) \leq \mathcal{H}(\mu_2)$ where $\mathcal{H}(\mu_1)$ takes values within 0.01 of those shown on the $x$-axis of Fig. 6. Similarly, $\mu_2$ is sampled to evenly cover the range of allowed values $\mathcal{H}(\mu_2) \in [\mathcal{H}(\mu_1), 5]$. To construct distance matrices, we sample 32 points uniformly on $m_\mathcal{X} \in \{2, 8, 32\}$ dimensional spheres with radii 0.5 and take pairwise Euclidean distances. As seen in the

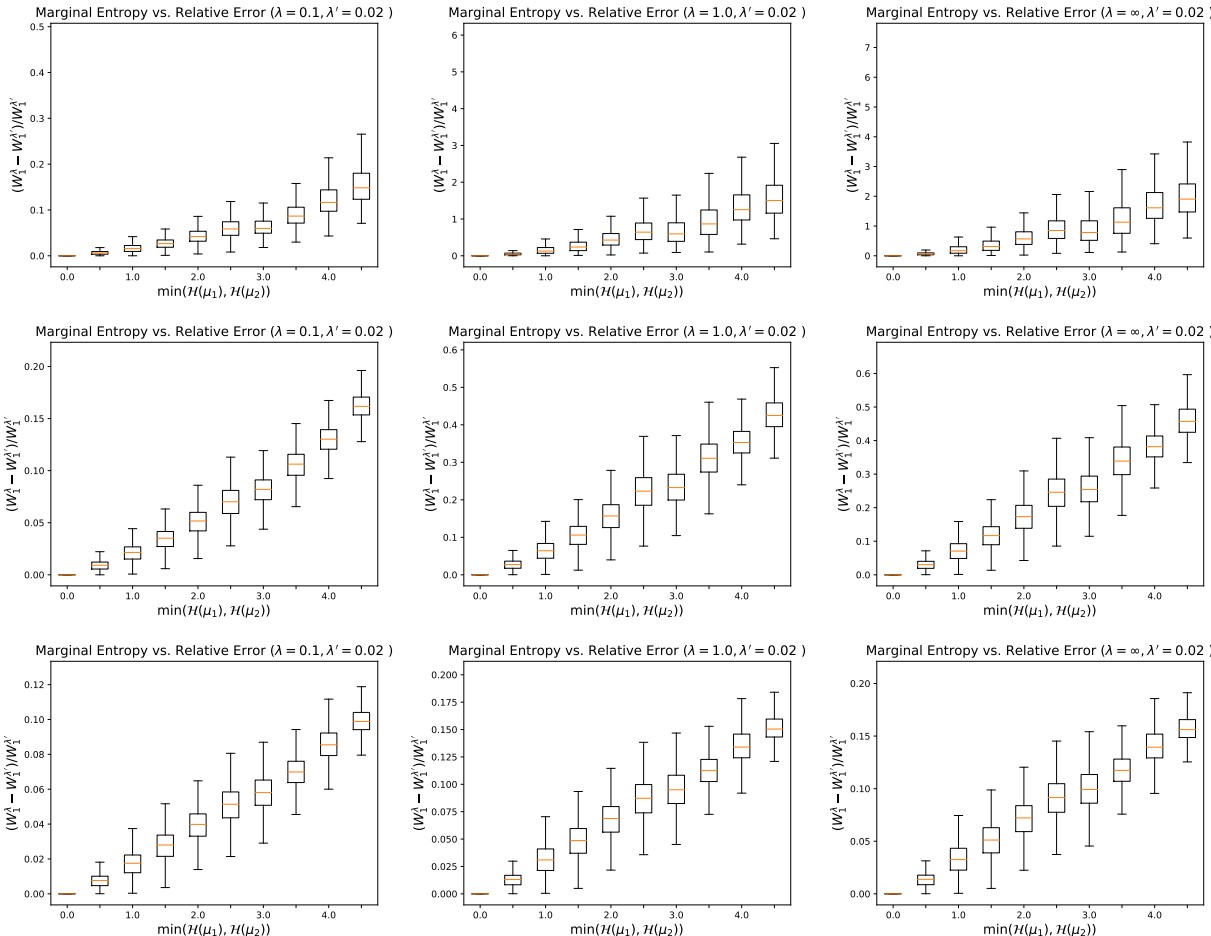

Figure 6: Measurements of relative error against minimum marginal entropy with varying $\lambda \in \{0.1, 1, \infty\}$ **(left-to-right)** and ambient space dimensionality $m_\mathcal{X} \in \{2, 8, 32\}$ **(top-to-bottom)**. Each individual box corresponds to 1000 pairs of $(\mu_1, \mu_2)$ with 20 and 50 samples of probability vectors $\mu_1$ and $\mu_2$.

bottom-left corner of all plots, we have strict equality $W_1^\lambda = W_1^{\lambda'}$ for all settings with $\mathcal{H}(\mu_1) = 0$ as predicted by (41). Furthermore, we observe a monotonic relationship between $\mathcal{H}(\mu_1)$ and the expected quality of weaker metrics $W_1^\lambda$ across all settings of $\lambda$ and $m_\mathcal{X}$, the latter of which controls the distribution of distance values.[8]

# D   Implementation Details

## D.1   Warm-starting Sinkhorn distance computation

Recall from Cuturi (2013) that by Sinkhorn's Theorem (Sinkhorn, 1967) the unique optimal transport plan $\omega^*$ for the entropy-regularized problem for a finite space with $\ell$ elements can be written in matrix form as $\mathrm{diag}(\boldsymbol{u}) K \mathrm{diag}(\boldsymbol{v})$, where $\boldsymbol{u}, \boldsymbol{v} \in \mathbb{R}_+^\ell$ and $K_{ij} = e^{-\lambda d(x_i, x_j)}$. Then, one iteratively updates vectors $\boldsymbol{u}$ and $\boldsymbol{v}$ in alternation so as to satisfy the marginal constraints on row and column sums. The standard implementation of this algorithm in the Python Optimal Transport package (Flamary et al., 2021) initializes $\boldsymbol{u}$ and $\boldsymbol{v}$ to be $\boldsymbol{1}_\ell / \ell$. In our case, the sequence of Sinkhorn problems being solved follow a structure: (i) for each of $n$ fixed-point updates $\mathcal{F}_\pi(\widehat{d}_\pi)$ we compute $W_p^\zeta(\widehat{d}_\pi)(\mathcal{P}_\pi(\boldsymbol{s}_i), \mathcal{P}_\pi(\boldsymbol{s}_j))$ for all state pairs $(\boldsymbol{s}_i, \boldsymbol{s}_j)$, and (ii) after $n$ updates, we update the policy $\pi$. Since $\mathcal{F}_\pi$ is contractive, the consecutive metrics approach one another as we apply the mapping $\mathcal{F}_\pi$. Thus, the solutions shall also approach one another considering Lemma 3.2.

---

[8]The distribution of pairwise distances for uniformly sampled points on a unit $n$-sphere approximately follows $\mathcal{N}(\sqrt{2}, \frac{1}{2n})$ (Wu et al., 2017). In our case, the distribution becomes increasingly concentrated around $\sqrt{2}/2$ with increasing $m_\mathcal{X}$.

---

**Algorithm 1** $\epsilon$-aggregation for finite $\mathcal{S}$

---

**Require:** Distance matrix $D \in \mathbb{R}_+^{|\mathcal{S}|^2}$, aggregation threshold $\epsilon \geq 0$

  `num_neighbours` $\leftarrow \Sigma_j \mathbf{1}[D_{ij} \leq \epsilon]$

  `partitions` $\leftarrow [\,]$

  `assigned` $\leftarrow \text{zeros}(|\mathcal{S}|)$

  **while** $\sum_i \texttt{assigned}[i] < |\mathcal{S}|$ **do**

    $m \leftarrow \arg\max_i(\texttt{num\_neighbours})$                         ▷ Greedily select medioid index.

    $\texttt{assigned}[m] \leftarrow 1$

    `members` $\leftarrow \text{argwhere}_j(D_{mj} \leq \epsilon \text{ and not } \texttt{assigned}[j])$

    `partition` $\leftarrow [m] + \texttt{members}$                   ▷ List concatenation.

    $\texttt{assigned}[\texttt{members}] \leftarrow 1$          ▷ All new members of the partition marked assigned.

    $\texttt{num\_neighbours}[\texttt{partition}] \leftarrow -\infty$     ▷ Assigned elements should not be selected as medioids.

    `partitions` $\leftarrow \texttt{partitions} + [\texttt{partition}]$       ▷ Append partition to partitions.

  **end while**

  $|\widetilde{\mathcal{S}}| \leftarrow \text{length}(\texttt{partitions})$

  $\Phi \leftarrow \text{zeros}(|\mathcal{S}| \times |\widetilde{\mathcal{S}}|)$

  **for** $j$ in $[1, \ldots, |\widetilde{\mathcal{S}}|]$ **do**

    **for** $i$ in $\texttt{partitions}[j]$ **do**

      $\Phi_{ij} \leftarrow 1$

    **end for**

  **end for**

  **return** $\Phi \in \{0,1\}^{|\mathcal{S}| \times |\widetilde{\mathcal{S}}|}$

---

A similar observation can be made about small policy updates, which likely change the fixed-point metric only slightly (e.g., see Lemma 3.6). Inspired by Ferns et al. (2006), we exploit this structure by saving the final vectors $\boldsymbol{u}_{ij}$ and $\boldsymbol{v}_{ij}$ for all state pairs $(\boldsymbol{s}_i, \boldsymbol{s}_j)$ and initializing each run of the Sinkhorn-Knopp algorithm with the most recently saved $(\boldsymbol{u}_{ij}, \boldsymbol{v}_{ij})$ for the corresponding pair of states. This results in up to an order of magnitude improvement in wall-clock time as shown in Fig 5.

### D.2   An algorithm for hard aggregation for finite $\mathcal{S}$

In Algorithm 1, we provide the simple greedy algorithm that we used for partitioning a finite space $\mathcal{S}$ given a pairwise distance matrix and a threshold $\epsilon$. The algorithm counts for each state the number $\epsilon$-close states and greedily assigns partition medioids based on this simple heuristic. Each partition contains a medioid and its $\epsilon$-neighbours which have not been previously assigned to another partition. While the algorithm itself is not necessarily optimal, we showed empirically in Figs. 1 and 3 that with a good enough metric it recovers the ground-truth partitions and is therefore sufficient for our purposes.

## E   Bisimulation Distance vs. Absolute Value Difference

In Figs. 1 and 3, we measured $\max_{\boldsymbol{s}, \boldsymbol{s}'} \left| \widehat{d}_\pi(\boldsymbol{s}, \boldsymbol{s}') - |V^\pi(\boldsymbol{s}) - V^\pi(\boldsymbol{s}')| \right|$ over time-steps $k$ as a proxy for VFA capabilities of the running bisimulation metric. While the true $\pi$-bisimulation metric $d_\pi^\sim$ provably satisfies $\Delta V^\pi(\boldsymbol{s}, \boldsymbol{s}') = |V^\pi(\boldsymbol{s}) - V^\pi(\boldsymbol{s}')| \leq d_\pi^\sim(\boldsymbol{s}, \boldsymbol{s}')$, we only approximate it with $n$ fixed-point updates after each policy update. As such, $\widehat{d}_\pi$ may under-estimate $\Delta V^\pi(\boldsymbol{s}, \boldsymbol{s}')$ especially early in training since we initialize $d_0 = \mathbf{0}$. In Fig. 7, we show box plots of $\widehat{d}_\pi(\boldsymbol{s}, \boldsymbol{s}') - \Delta V^\pi(\boldsymbol{s}, \boldsymbol{s}')$ over time to investigate this behavior. In particular, we run the API($\alpha_k$) algorithm on the first MDP with $\lambda \in \{0.25, 2.0, \infty\}$, $n \in \{1, 5\}$, $\epsilon = 0.1$ and $\alpha_k = \max(0.01, k^{-0.8})$. We only run the algorithm for 200 steps here since the metric stabilizes by then. We observe that for $n = 1$, the approximate metric $\widehat{d}_\pi$ starts to over-estimate $\Delta V^\pi(\boldsymbol{s}, \boldsymbol{s}')$ within the first 30 steps as the metric approaches the fixed-point metric $d_\pi^\sim$. In contrast, when $n = 5$ a single API step is sufficient for $\widehat{d}_\pi$ to exceed $\Delta V^\pi(\boldsymbol{s}, \boldsymbol{s}')$. As expected, we find a weaker upper bound on $\Delta V^\pi(\boldsymbol{s}, \boldsymbol{s}')$ with increasing $\lambda$.

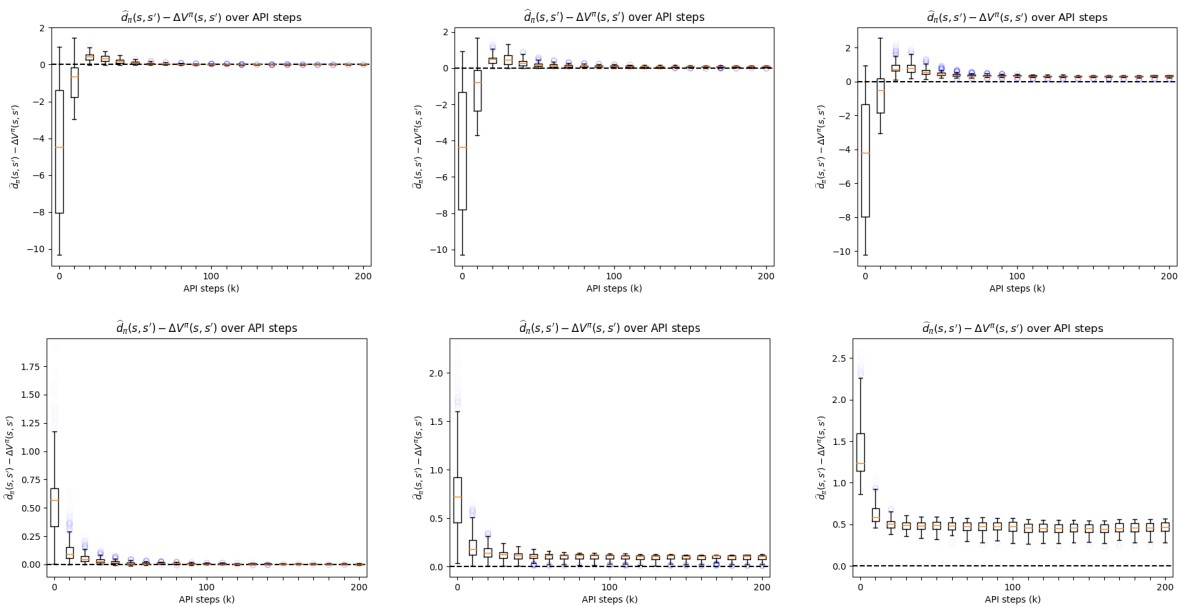

Figure 7: Approximate bisimulation distance vs. the absolute value difference over time for $n = 1$ **(Top)** and $n = 5$ **(Bottom)** with varying $\lambda \in \{0.25, 2.0, \infty\}$ **(left-to-right)**.

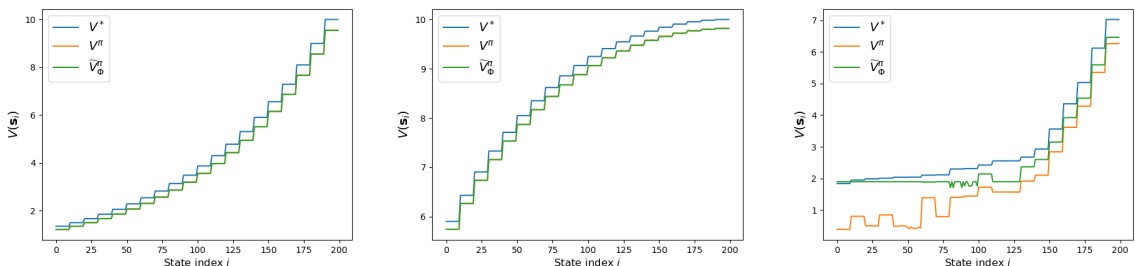

Figure 8: Value functions for the first and second MDPs from Sec. 4 **(Left, Middle)** and a randomly generated MDP from Appendix F **(Right)** after 1000 steps of training with $\lambda = \infty$ and 30-medioids partitioning.

## F An Additional Class of MDPs

In this section, we repeat some of the main experiments in Sec. 4 for a new class of randomly generated MDPs. As before, we have $m = 20$ equivalence classes (ECs) and 200 states. Each equivalence class $B_i$ is assigned an optimal action $a_j \in \mathcal{A}$ where $|\mathcal{A}| = 10$ and $j = i \% 10$. If the agent takes the optimal action in $B_i$, the MDP transitions to $B_{i+1}$ with probability $1 - p_i$, where $p_i \sim \mathcal{U}(0, 0.25)$, except in $B_m$ the agent stays in $B_m$ with probability $1 - p_m$. With probability $p_i$, the agent transitions to a randomly selected EC other than $B_i$ and $B_{i+1}$. Taking any of the non-optimal $|\mathcal{A}| - 1$ actions transition the agent back to $B_1$ from any $B_i$ with probability 1. As before, transition probabilities from a state $s_i$ to a given EC are sampled uniformly from the $(|\mathcal{S}|/m - 1)$-simplex. The agent collects a unit reward only when it takes the optimal action in $B_m$.

In Fig. 8, we illustrate the value functions corresponding to the MDPs analyzed in Sec. 4 and the MDP discussed here; the approach here with randomly sampled transitions results in irregular step sizes between ECs for $V^*$, although value equivalence is preserved within each EC. The task is rendered more difficult since many states in different ECs have similar values and the importance of more precise metrics for VFA is highlighted. Indeed, we observe that $\pi$ significantly underperforms the optimal policy in this case and the bisimulation-based approximation $\widetilde{V}_\Phi^\pi$ of $V^\pi$ has high error particularly for ECs with a small value difference.

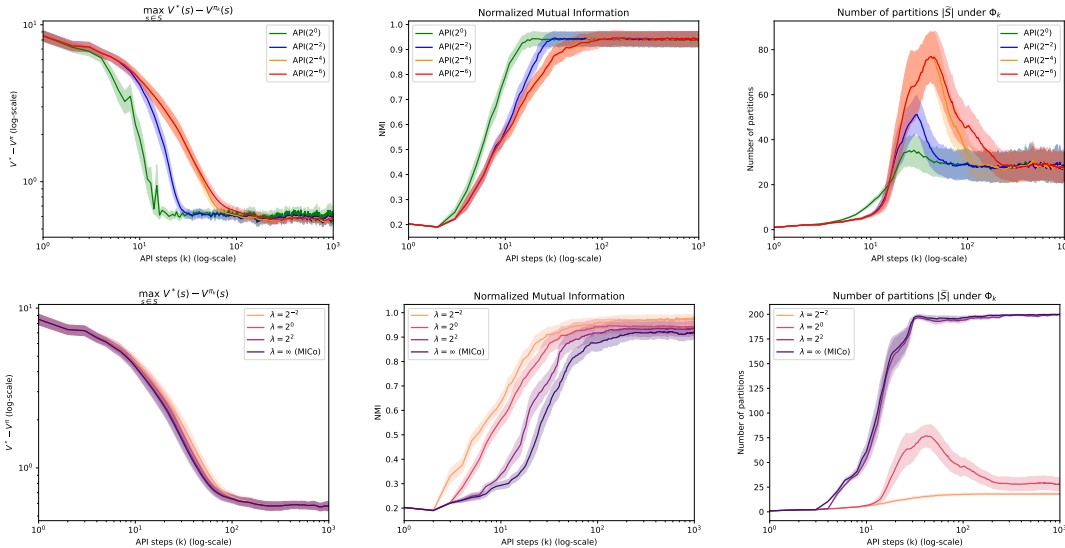

Figure 9: Ablations of $\alpha$ **(Top)** and $\lambda$ **(Bottom)** for the third class of MDPs introduced in Appendix F.

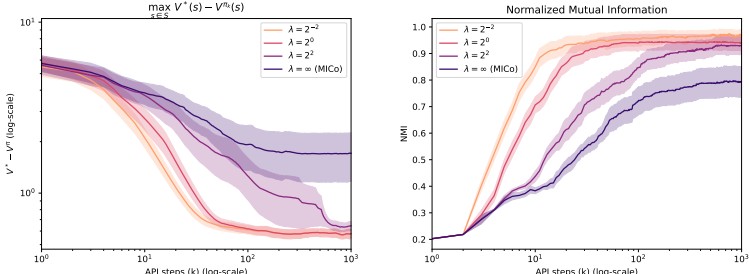

Figure 10: Ablation of $\lambda$ for the new class of MDPs in the limited representation capacity setting, where Algorithm 1 is replaced with 30-medioids partitioning.

In Fig. 9, we ablate $\alpha$ and $\lambda$ for this new class of MDPs and observe similar results to Sec. 4. Namely, we note better asymptotic performance but slowed down progress with smaller $\alpha$. While the setting $\lambda$ does not influence performance, lower $\lambda$ produces better metrics and a more efficient partitioning. Lastly, in Fig. 10, we repeat the experiment shown in Fig. 4 where the number of allowed partitions is limited to 30. We again obtain qualitatively similar results with lower $\lambda$ producing a higher quality metric and a better policy.

