# OpenReview forum: "Approximate Policy Iteration with Bisimulation Metrics"
_TMLR — Accepted by TMLR_

### Review · Reviewer_7DzN · 2022-08-23

**Summary Of Contributions:**

This paper reformulates bisimulation and $\pi$-bisimulation metrics in a more general form via Sinkhorn distances. The authors demonstrate that this reformulation recovers traditional bisimulation metrics as well as a recent behavioural metric (MICo, from Castro et al., 2021) as extreme points of one of the parameters for Sinkhorn distances.

Using this more general form, the authors define an approximate policy iteration procedure which uses these metrics for state aggregation; they prove performance bounds, as well as convergence results when improving policies and metric approximations concurrently.

Finally, the authors present empirical validations of their theoretical work on a series of controlled experiments.

**Broader Impact Concerns:**

N/A.

**Requested Changes:**

See suggestions above.

**Strengths And Weaknesses:**

I found this paper to be very well-motivated and well-written. The authors are building on prior work in bisimulation-based metrics and nicely extend existing results in a more generalized manner. I particularly appreciated the finding that, under the Sinkhorn formulation with parameters $(p, \zeta)$, one can recover bisimulation metrics with $(p, \zeta) = (1, 0)$ and MICo with $(p, \zeta) = (1, \infty)$.

In general, I found the discussion around their theoretical results very useful, and helped highlight their significance.

I found the empirical investigations well-structured, and effective at validating the insights from the theoretical sections.

I found little to fault, but here are a few minor suggestions:

1. It would be nice to use hyperref throughout so that the reader can click through the citations and equation/figure references.
2. Remove the period after "_compact_" in the the third line of section 2.1
3. In section 3 when discussing "policy oscillation", this recent work could be of interest: https://arxiv.org/abs/2206.00730
4. Minor nit in page 8, where DBC and MICo are listed as "such actor-critic approaches": neither are limited to actor-critic methods and can be applied to value-based methods (in fact, most of MICo's experiments were on value-based methods).
5. It would be useful to specify what is being measured in each row in Figures 2 and 3.
6. The first sentece of section 4 would read better as "In this section, we conduct experiments to _empirically_ investigate the ... results in Sec. 3."
7. In section 4 I would choose a different word than "booted", which sounds too informal. Just "transition" maybe?

---

> ### Author Response · Authors · 2022-10-12
> **Thank you for your comments**
>
> We thank the reviewer for their positive remarks on our paper. We are pleased to hear that the reviewer found the paper to be “very well-motivated and well-written”, “discussion around theoretical results very useful”, “empirical investigations well-structured, and effective at validating the insights from the theoretical sections”, that the reviewer appreciated our unifying view of MICo and bisimulation via Sinkhorn distances, and that they “found little to fault”.
>
> Regarding minor suggestions, we have incorporated all the requested changes (by fixing typos and re-wording statements) into the new revision with the exception of #3. Upon taking a close look at the provided reference on “policy churn”, we found that in Sec. 5 authors note: “A phenomenon in the literature that is related, *but not equivalent*, to policy churn is that of policy oscillation or chattering” and that “[they] do not think policy oscillation is a likely explanation for policy churn”. Seeing as the authors note that the relevance is thought to be only tangential, we opted not to include this reference for this revision. However, if the reviewer deems necessary, we can add a citation for this work.

---

### Review · Reviewer_PeHo · 2022-08-26

**Summary Of Contributions:**

This paper makes two main conceptual contributions:

First, the authors give a generalization of bisimulation metrics, which is defined using the 1-Wasserstein distance, to the cases of p-Wasserstein and Sinkhorn distances. The benefits for doing so are due to improved computational costs (S^2log(S) instead of S^3log(S)).

Second, the authors propose an approximate policy iteration (API) method that makes use of \pi-bisimulation to conduct a form of adaptive state aggregation. The essence of the method is that on each iteration, it first uses the current policy \pi_k to compute the \pi_k-bisimulation (using n iterations to compute the fixed point), and then applies the evaluation and improvement steps on an aggregated state space implied by the bisimulation metric.

The other contributions are:

An extension of the new API algorithm that makes smaller / smoothed updates (specified using a parameter \alpha). This is based on the API(\alpha) algorithm from Bertsekas, 2018.

Empirical investigations: (1) of \alpha, showing that setting it to a large value initially and then decreasing it with iteration is a promising strategy, (2) of the parameters used in the Sinkhorn distance, (3) walltimes for different values of \alpha, resulting in the observation that smaller \alpha means similar bisimulation metrics and therefore higher value to warm-starting.

**Requested Changes:**

The numbers below reference the weaknesses section.

[1] Critical
[2] Would strengthen
[3] Would strengthen
[4] Would strengthen

**Strengths And Weaknesses:**

Strengths:

I find the conceptual contributions of the paper to be interesting, creative, and relevant to the community. The paper is very well-written and organized, giving clear explanations and detailed, yet easy-to-follow theoretical analysis. I read through most of the proofs and did not find any major issues; the paper extends and generalizes classical theory from the ADP literature (e.g., those found in Bertsekas textbooks) in an elegant fashion.

Weaknesses:

[1] There could be a discussion to other forms of adaptive aggregation that are not necessarily based on bisimulation. There was little discussion in this direction in the literature review and as a reader, I would’ve liked to get a better understanding of when to apply certain methods. A recent example is https://arxiv.org/abs/2107.11053.

[2] The empirical results use \gamma = 0.9. My guess is that the theoretical bounds become impractical as \gamma -> 1 due to the (1-\gamma)^(3 or 4) terms in the results, but it would be nice to know how the empirics change with high \gamma.

[3] There is little discussion on where/when to apply this algorithm vs other types of aggregation or approximate DP / RL approaches and there is no comparison to any baseline algorithms (with the exception of, perhaps, MICo). It makes it hard for the reader to gauge when this approach is practically useful. Note that I am not advocating for a gigantic empirical section with comparisons to everything, but there should at least be a few naive baselines.

[4] Discussion on how to choose $n$ and the case of default API (n=0?). The practical question becomes: given a fixed amount of computation, how much effort should go toward the bisimulation part and how much effort toward the API part? One special case, at least for the finite state / action case, would be no aggregation and just applying default API.

---

> ### Author Response · Authors · 2022-10-12
> **Thank you for your time and comments**
>
> We thank the reviewer for their positive remarks including those on the writing and organization of our paper, creativity and allure of its concepts, the relevance of its contributions to the community, and the elegance of its generalized ADP results. We also thank the reviewer for their constructive feedback aimed at strengthening the paper. Below, we respond to the reviewer's feedback.
>
> [1] We thank the reviewer for pointing out this recent reference. We had strived to focus the scope of our discussion to bisimulation metrics and their generalizations. However, we understand the reviewer's concern and have thus added a new section Appendix B solely for further discussion of other state aggregation methods in the literature. This new section is pointed to at the end of Sec 2.1 in the new revision. The new section includes the reference recommended above along with others. We note a key distinction of our algorithm from that of the listed reference: the online algorithm proposed by Chen et al. (2021) requires exact updates over the original state space (albeit infrequently), while our algorithm performs VFA strictly over reduced state spaces.
>
> [2] Following the reviewer's comments we repeated the experiments shown in Fig. 2 with $\gamma = 0.99$ and did not see any qualitative differences. We added this comment to the discussion of Fig. 2 in the main text.
>
> [3] First, we note that the setting $p=2$ in Fig. 3 is comparable to the DBC formulation (for small $\lambda$) since DBC uses the $2$-Wasserstein metric in practice (for fast computation over Gaussian latent dynamics models). Hence, in principle, the $(p, \lambda)$ Sinkhorn ablations in Fig. 3 compare against two metric formulations based on the DBC and MICO papers. We have revised Fig. 3 bottom row legends to add "(DBC)" next to "$p=2$" to clarify this point.
>
> Secondly, we note that the particular algorithm we introduced here is not necessarily intended for practical use. The algorithm is designed to mimic bisimulation-based actor-critic approaches (as discussed in Sec 3.3), but with VFA via piecewise constant functions (for ease of analysis) rather than arbitrarily architectured, trained and tuned neural networks. Beyond being shown to be GPU-friendly and feasible for finite state spaces with $|\mathcal{S}|=200$, the API algorithm chiefly serves as a tool to perform a strictly controlled analysis (both mathematical and empirical) of different design choices when using bisimulation metrics in RL (such as the choices of $p, \lambda$ and $\alpha$).
>
> Regarding a discussion of when to apply bisimulation or aggregation approaches in general, we quote the last paragraph of Sec. 2.2: *"if substantial reductions in the size of the state space are possible (e.g., due to the presence of distractors) such that $|\mathcal{S}| \gg |\widetilde{\mathcal{S}}|$*, one can pre-compute bisimulation-based abstractions to find a near-optimal policy much more quickly. However, computing the metric itself exactly can be costly. ... Hence, fast approximations of bisimulation metrics are necessary in practice to amortize the cost of metric learning and enable usage in continuous MDPs or large finite MDPs." Indeed, speeding up the computation of bisimulation metrics has been a key focus of our paper, namely via the reduction in complexity from $\tilde{O}(\mathcal{S}^5)$ to $\tilde{O}(\mathcal{S}^4)$ using the Sinkhorn formulation.
>
> [4] Regarding the choice of $n$, we refer the reviewer to Fig. 2 for an ablation of $n$ and its discussion in Sec. 4 (beginning of Page 10). We reported that when warm-starting is used, smaller n only slightly delays the performance improvement of the policy. This is also true for the smallest possible value of $n=1$. Hence, in the warm-start case smaller n offers a favorable trade-off in terms of wall-clock time. On the other hand, for the naive algorithm in Thm. 3.5, one requires high $n$ to ensure an accurate enough metric at each iteration, which results in the policy oscillating near optimality rather than stalling far from it as it does for small $n$. Appendix D and Fig. 7 also discuss how different $n$ influences the metric's approximation accuracy of the absolute value difference.
>
> For $n=0$, the metric would stay at its initial value of $0$ for all state pairs so that all states would collapse to a single abstract state (partition). This would imply poor VFA and little-to-no progress in terms of policy performance, hence the case $n=0$ was omitted.
>
> Regarding a comparison to default API, we highlight that the algorithm we used for analysis is intended to illustrate and validate components of the theory rather than being a competitive algorithm for tractably-sized finite MDPs. One avenue for future work is replacing the exact fixed-point updates for learning the bisimulation metric with function approximation, in which case the API algorithm or its variants become practical for continuous spaces, and our theoretical analysis remains relevant.

---

### Review · Reviewer_wUnf · 2022-09-29

**Summary Of Contributions:**

This paper introduces a general family of bisimulation metrics, based on Sinkhorn distances. Sinkhorn distances generalize Wasserstein distances through entropy regularization, and come with certain benefits such as faster convergence. The paper then builds an approximate policy iteration algorithm that uses state aggregation based on the previously introduced bisimulation metrics. The authors then extend this algorithm to a conservative policy iteration variant, which is useful when warm starting the distance metrics. This is followed by empirical analyses on small, finite state MDPs (<500 states) to confirm the faster convergence benefits of using Sinkhorn distances as well as stability benefits offered by the conservative policy iteration version.

**Broader Impact Concerns:**

I do not think the paper's contributions directly cause any ethical concerns.

**Requested Changes:**

I would like to have seen some experiments on pixel-based domains with changes inspired by the different distance metrics and using conservative policy gradient algorithms such as TRPO/MDPO. Having said that, I think the paper in its current form is still complete and warrants a clear acceptance.

**Strengths And Weaknesses:**

Strengths
- Overall the paper is very well structured. The writting is extremely clear the authors do a good job of explaining each step before diving into the mathematical details. The experiments are done rigorously and they accompany the algorithmic contributions well. Moreover, using conservative updates along with an analysis on different distance metrics is an important direction for improving practical algorithms, and the paper does a good job at making this connection clear.

Weaknesses
- Extensions to API($\alpha$) seem straightforward, as do the convergence proofs for the Sinkhorn Bisimulation versions. Therefore, I would not cast the paper as doing something super novel. But maybe that's not the aim of the paper as well and I think that does come across well. More importantly, I would have liked to see the benefits of better distance metrics as well as conservative updates for neural network architectures, since a large portion of the paper is inspired by DBC and MiCO.

*Just wanted to note that DBC has been recently shown to not reproducible in many recent papers, and seems to have very poor performance for most tasks. I note this only because the authors mention the connections to DBC quite often.

---

> ### Author Response · Authors · 2022-10-12
> **Thank you for your feedback**
>
> We thank the reviewer for their feedback noting that “the paper is very well structured”, “the writing is extremely clear”, “experiments are done rigorously and they accompany the algorithmic contributions well” and “the paper in its current form is still complete and warrants a clear acceptance”. As also noted by the reviewer, we indeed strived to provide theoretical and empirical insights on bisimulation metrics and API rather than seek algorithmic novelty. We agree with the reviewer that empirical investigations of the insights provided in this paper in relation to neural architectures (e.g., via TRPO or MDPO) and evaluation on pixel-based domains are important directions for future research; however, we left this outside the scope of the current paper for reasons mentioned at the end of Sec. 3.

---

### Decision · Action_Editors · 2022-11-04

**Recommendation:** Accept as is

**Comment:**

All reviewers vote for a clear accept, while any concerns with the original submission have been addressed in a revision in the meantime, so that I recommend to accept the paper in its current form.

**Audience:**

The paper deals with a common topic in the RL literature and presents some new findings that will be of interest to at least a subset of the RL community.

**Claims And Evidence:**

The claims of the submission are supported by theoretical results as well as experiments. Some issues with respect to the latter brought up in the reviews have been addressed in an update of the paper.